

# Wintertime photochemistry in Beijing: Observations of ROx radical concentrations in the North China Plain during the BEST-ONE campaign

Zhaofeng Tan[1,2], Franz Rohrer[2], Keding Lu[1], Xuefei Ma[1], Birger Bohn[2], Sebastian Broch[2],
Huabin Dong[1], Hendrik Fuchs[2], Georgios I. Gkatzelis[2], Andreas Hofzumahaus[2], Frank Holland[2],
Xin Li[1], Ying Liu[1], Yuhan Liu[1], Anna Novelli[2], Min Shao[1], Haichao Wang[1], Yusheng Wu[1†],
Limin Zeng[1], Min Hu[1,3], Astrid Kiendler-Scharr[2], Andreas Wahner[2], Yuanhang Zhang[1,3]

[1] College of Environmental Sciences and Engineering, Peking University, Beijing, 100871, China
[2] Institute of Energy and Climate Research, IEK-8: Troposphere, Forschungszentrum Jülich GmbH, Jülich, Germany
[3] Beijing Innovation Center for Engineering Sciences and Advanced Technology, Peking University, 100871, Beijing, China
[†] Now at Department of Physics, University of Helsinki, Helsinki, Finland.

**Correspondence to:** Keding Lu (k.lu@pku.edu.cn) and Yuanhang Zhang (yhzhang@pku.edu.cn)

**Abstract.** The first wintertime in-situ measurements of hydroxyl (OH), hydroperoxy (HO$_2$) and organic peroxy (RO$_2$) radicals (RO$_x$=OH+HO$_2$+RO$_2$) in combination with observations of total reactivity of OH radicals, $k_{OH}$ in Beijing are presented. The field campaign "Beijing winter finE particle STudy - Oxidation, Nucleation and light Extinctions" (BEST-ONE) was conducted at the suburban site Huairou near Beijing from January to March 2016. It aimed to understand oxidative capacity during wintertime and to elucidate the secondary pollutants formation mechanism in the North China Plain (NCP). OH radical concentrations at noontime ranged from 2.4×10$^6$ cm$^{-3}$ in severely polluted air ($k_{OH}$ ~ 27 s$^{-1}$) to 3.6×10$^6$ cm$^{-3}$ in relatively clean air ($k_{OH}$ ~ 5 s$^{-1}$). These values are nearly two-fold larger than OH concentrations observed in previous winter campaign in Birmingham, Tokyo, and New York City. During this campaign, the total primary production rate of RO$_x$ radicals was dominated by the photolysis of nitrous acid accounting for 46% of the identified primary production pathways for RO$_x$ radicals. Other important radical sources were alkene ozonolysis (28%) and photolysis of oxygenated organic compounds (24%). A box model was used to simulate the OH, HO$_2$ and RO$_2$ concentrations based on the observations of their long-lived precursors. The model was capable of reproducing the observed diurnal variation of the OH and peroxy radicals during clean days with a factor of 1.5. However, it largely underestimated HO$_2$ and RO$_2$ concentrations by factors up to 5 during pollution episodes. The HO$_2$ and RO$_2$ observed-to-modeled ratios increased with increasing NO concentrations, indicating a deficit in our understanding of the gas-phase chemistry in the high NO$_x$ regime. The OH concentrations observed in the presence of large OH reactivities indicate that atmospheric trace gas oxidation by photochemical processes can be highly effective even during wintertime, thereby facilitating the vigorous formation of secondary pollutants.

## 1 Introduction

After a series of air pollution control provisions have been implemented by the Chinese government to improve the air quality in China, emissions and concentrations of primary pollutants (sulfur dioxide, nitrogen oxide, and coarse





particulate) have been showing a fast decreasing trend national wide in recent years. However, secondary pollution characterized by high concentrations of ozone and fine particles $PM_{2.5}$ has now become the major concern. Especially, severe pollution events (haze) frequently happened in winter of the last few years, with very poor visibility and extreme high concentrations of $PM_{2.5}$, secondary aerosol and its precursors (Sun et al., 2006; Yang et

al., 2011; Guo et al. 2014; Zhao et al., 2013).

The OH radical is the major atmospheric oxidizing agent during the daytime, converting primary pollutants to secondary ones. OH radicals attack pollutants, e.g. CO, non-methane hydrocarbons (NMHCs), producing peroxy radicals ($HO_2+RO_2$), which can regenerate OH by their reaction with NO. Within this $RO_x$ cycling, primary pollutants are converted to secondary pollutants, e.g. to $CO_2$, $H_2SO_4$, $HNO_3$, oxygenated organic compounds

(OVOCs) etc., some of which are important precursors for aerosol. In addition, NO is oxidized to $NO_2$ in the reaction with peroxy radicals leading to net ozone production and NO regeneration via further photolysis. The efficient coupling of the $RO_x$ and the $NO_x$ cycles guarantees the fast degradation of primary pollutants with the disadvantage of the formation of secondary pollutants. In wintertime, the radical chemistry is less active than summertime because the dominating primary source of OH radicals, photolysis of ozone, is strongly reduced by the smaller photolysis rate

and the lower water vapor abundances at low temperatures during wintertime. Global models predict OH concentrations to be on average only $0.4 \times 10^6 \, cm^{-3}$ in northern China in January, one order of magnitude lower than what is predicted for summertime for the lower troposphere in northern China (Lelieveld et al., 2016), which is consistent with the general expectation. However, composition analysis showed that the contribution of secondary components to the aerosol increase during pollution events in the NCP (Guo et al., 2014; Zhao et al., 2013; Huang et

al., 2014), which suggests that oxidation plays an important role in aerosol formation. However, a rigorous test of the current understanding of radical chemistry by direct in-situ measurements of ambient OH concentration is missing during wintertime in China. Therefore, the role of radicals in air pollution is unclear for wintertime, especially during heavy pollution episodes in China.

Only a handful field campaigns have been carried out in northern China to elucidate key processes of tropospheric

radical chemistry. In summer 2006, a comprehensive campaign took place at a suburban site in Beijing (Yufa), 40 km south of the center of the city. High concentrations of OH radicals were observed. In addition, the measured OH reactivity showed a high concentration of reactive trace gases. The combination of high OH concentration and high OH reactivity resulted in a fast formation of secondary pollutants. Box model calculations could not reproduce the ambient OH concentrations for the low $NO_x$ regime (Lu et al., 2013). In summer 2014, another field campaign was

conducted at a rural site in the center of the NCP (Wangdu) to study the formation of secondary pollutants (Tan et al., 2017; Fuchs et al., 2017b). Also for this environment, there is a tendency of underestimating OH for conditions when NO was less than 300 pptv. Additionally, a missing peroxy radical source was found for the high $NO_x$ regime showing a severe underestimation of local ozone production (Tan et al., 2017).

To reveal the cause of heavy air pollution episodes and the link between radical chemistry and air pollution during

wintertime, a field campaign "Beijing winter finE particle STudy - Oxidation, Nucleation and light Extinctions" (BEST-ONE) was carried out at the suburban site (Huairou) in the Beijing area from January to March 2016. For the first time, concentrations of OH, $HO_2$, $RO_2$ radicals and of the reactivity of OH, $k_{OH}$, were simultaneously measured in China during wintertime in addition to numerous measurements of other trace gas concentrations and aerosol



properties. In this study, the ambient radical concentrations are analyzed for different chemical conditions by calculating the budget of radicals and by comparing measurements to model calculations.

## 2 Experimental

### 2.1 Location

The BEST-ONE campaign took place at the suburban site Huairou (40.41°N, 116.68°E), 60 km northeast to the center of Beijing. Instruments were set-up on the campus of the University of the Chinese Academy of Sciences (UCAS) as shown in Fig. 1. Beijing is located on the northwest edge of the NCP and is confined by the Yanshan Mountains from the northeast to the west. The topography of Beijing facilitates the accumulation of pollutants if southern winds carry emissions from the NCP region. The measurement site was located in a valley of the Yanshan

Mountains, acting as an intermediary of air mass transportation from downtown Beijing. A road with moderate traffic and a small village were located east of the university campus. Thus, local anthropogenic activities might have influenced the measurement site, especially during traffic rush hours and cooking hours.

Most instruments were placed on the top floor of a 20 meters high building with their inlet lines going through the ceiling. All inlet lines were set at the same height as close as possible to each other to guarantee to sample of the

same air masses. The sampling height for all inlet lines was 1.7 meters above the surface of the building.

### 2.2 Instrumentation

#### 2.2.1 OH, HO$_2$, and RO$_2$ measurements

OH was measured by a laser-induced fluorescence system. HO$_2$ and RO$_2$ concentrations were detected as OH after chemical conversion by adding excess NO in another two separated cells. Detailed information of this instrument can

be found in the paper by Tan et al. (2017) and the references therein. Only a brief description of the radical instrument is presented here.

Ambient air was drawn into a low-pressure cell (4 hPa) through a nozzle with 0.4 mm diameter pinhole at a rate of 1 standard liter per minute (SLM) to measure OH concentrations. The OH radicals were excited by a pulsed laser at 308 nm (repetition rate: 8.5 kHz; pulse duration at full width half maximum (FWHM) 25 ns, laser power 20 mW).

Subsequent fluorescence photons were collected by a lens system and detected by a Multiple Channel Plate detector (Photek, PMT 325).

HO$_2$ was converted to OH by NO addition below the sample nozzle in a second fluorescence cell that had otherwise the same design as the OH cell. It is known that the measurement of HO$_2$ by chemical conversion can introduce an interference from specific RO$_2$ radicals (Fuchs et al., 2011; Whalley et al., 2013; M. Lew et al., 2018). In this study,

the NO addition was reduced to minimize the RO$_2$ interference resulting in HO$_2$ conversion efficiency smaller than 25%. The concentration of added NO was switched every two minutes between 2.5 ppmv and 10 ppmv of NO in a sample flow of 1 SLM and an N$_2$-sheath flow of 1 SLM. The HO$_2$ conversion efficiency was 5% and 25% with 2.5 ppmv and 10 ppmv of NO addition, respectively. No significant difference was found for the two HO$_2$ data sets showing that the HO$_2$ measurements were interference-free.



RO$_2$ measurements with the LIF instrument require the conversion of RO$_2$ to OH. This was done in two steps. In a reaction flow tube (pressure 25 hPa, volume 2.8 L, sample flow 7.5 SLM) high concentrations of CO (1100 ppm) and NO (0.7 ppmv) were added to convert RO$_x$ to HO$_2$ (Fuchs et al., 2008). In the second step, the HO$_2$ radicals were finally converted to OH radicals in fluorescence cell at lower pressure (4 hPa) using a flow of 5 sccm of pure NO

into a sample flow of 3.5 SLM and an N$_2$-sheath flow of 1 SLM.

The instruments were calibrated about every three days using a calibration source which generated quantifiable RO$_x$ radical concentrations (Holland et al., 2003; Fuchs et al., 2008). No significant trend was found for all determined sensitivities during the campaign. Thus, averaged sensitivities were used to derive the observed radical concentrations. The accuracies take into account the uncertainty of the calibration source (10%, 1σ) and the 1σ

standard deviation of the variability of individual calibration sensitivities: 10%, 13% and 11% for OH, HO$_2$, and RO$_2$, respectively.

### 2.2.2 Investigation of possible OH interferences by chemical modulation experiments

The LIF technique used in this and previous work discriminates OH fluorescence signals from non-resonant background signals by periodically switching the laser wavelength between positions on- and off-resonance with

respect to an OH radical absorption line (Hofzumahaus et al., 1996). This method cannot discriminate between OH sampled in ambient air and OH produced artificially in the detection cell. A long-known interference is the laser photolysis of ozone for which the ambient OH measurements are routinely corrected (e.g., Holland et al., 2003). This ozone interference is close to the detection limit of the current instrument. Another recently discovered interference is produced by NO$_3$ radicals, which may cause detectable OH signals if NO$_3$ exceeds 10 pptv at night (Fuchs et al.,

2016). Recently, it has been observed that artificial OH is produced in some LIF OH instruments presumably from oxidation products of biogenic VOCs (Mao et al., 2012; Novelli et al., 2014; Rickly and Stevens, 2018). Following the concept proposed by Mao et al. (2012), we applied a prototype chemical-modulation reactor (CMR) in a campaign at Wangdu, China, in summer 2014, in order to test whether the OH LIF measurements were influenced by unknown artifacts. Unexplained OH signals equivalent to $(0.5 - 1.0) \times 10^6$ cm$^{-3}$ were found with a systematic 1σ error

of $0.5 \times 10^6$ cm$^{-3}$, which was less than 10% of the ambient OH concentrations at daytime (Tan et al., 2017).

In the present study, an improved version of the CMR device was used for some selected time periods during clean and polluted air conditions. The device consisted of a Teflon tube with an inner diameter of 1.0 cm and a length of 8.3 cm. About 20 slpm of ambient air was drawn through the tube by a blower, of which 1 slpm were sampled into the OH detection cell. In the current design, two small stainless steel tubes (outer diameter 1/16 inches) were

arranged at the entrance of the Teflon tube opposite to each other. Either propane diluted with nitrogen or pure nitrogen was mixed into the ambient air flow by these injectors. The propane serves as an OH scavenger, which removed ambient OH with a scavenging efficiency (ε) of 0.9 before the sampled air enters the LIF detection cell. Ambient OH signals (S$_{CM}$) were determined from the signals obtained either with the addition of propane (S$_{prop}$) or pure nitrogen (S$_{N2}$),

$$S_{CM} = (S_{N2} - S_{prop})/\varepsilon \qquad (1)$$




The $S_{CM}$ signals were then converted to OH concentrations. The required calibration and the determination of ε were performed by means of the OH calibration source before and after each chemical modulation experiment, showing a stable performance of the instrumental setup for all experiments.

For comparison, OH concentrations were calculated from the signal $S_{WM}$ which was obtained by wavelength

modulation and corrected for the known ozone interference ($S_{O3}$). The laser wavelength was switched between on-resonance ($S_{on}$) and off-resonance ($S_{off}$) modes while the CMR was operated with pure nitrogen.

$$S_{WM} = (S_{on}-S_{off}) - S_{O3} \qquad (2)$$

In total, four chemical modulation experiments were performed during this campaign: from January 9 to 10, from January 13 to 14, from January 25 to 26 and from February 29 to March 2. The OH instrument was switched between

wavelength and chemical modulation method on a regular basis every 2 minutes. In Fig. 2(a), the OH measurements with wavelength modulation and chemical modulation from January 13 to 14 are shown as an example. The deviation between the two methods is not significant and is caused most likely by the noise of the OH determination. The same result is found in the other three experiments. The good agreement between the two OH measurement methods is also shown in the correlation plot (Fig. 2(b)). The slope of this correlation is close to unity for the various

encountered chemical conditions. No intercept is found indicating no significant bias in the low concentration range, e.g. for nighttime measurements. Therefore, it was concluded that the OH radical measurements performed with the PKU-LIF instrument during this study were not influenced by a significant interference for conditions encountered during the chemical modulation experiments that were representative for the entire campaign.

### 2.2.3 Trace gas measurements

The instrumentation used in the present study was similar to the previous campaign in Wangdu (Tan et al., 2017). The gas-phase measurements are listed in Table 1 together with the respective instrument performance information. Total OH reactivity ($k_{OH}$) is defined as the pseudo first-order rate coefficient for OH loss calculated as the sum of all sink terms due to OH radical reactants $X_i$, depending on their bimolecular ambient concentrations [$X_i$] and their rate coefficients with OH. The observed $k_{OH}$ values utilized within this study were measured with an instrument based on

laser-photolysis – laser-induced fluorescence (LP-LIF) (Lou et al., 2010; Fuchs et al., 2017a; Fuchs et al., 2017b). The loss of OH in the instrument resulted in a zero loss rate of $(3.0\pm0.3)$ s$^{-1}$. This was tested by sampling pure synthetic air without OH reactants. The zero decay value was subtracted from decay times measured during ambient sampling. During the BEST-ONE campaign, the limit of detection was 0.6 s$^{-1}$ (2σ) at a time resolution of 90 s. In the presence of ambient NO higher than 20 ppbv, which was observed from time to time during the BEST-ONE

campaign, bi-exponential OH decays were observed resulting from OH recycling by the reaction of HO2 with NO. It is still possible to obtain a reliable $k_{OH}$ measurement by assigning the faster decay time to the OH reactivity but with an overall accuracy of 20% $\pm$ 0.7s$^{-1}$ compared to (5-10) % $\pm$ 0.7s$^{-1}$ for ambient NO concentrations < 20 ppbv (Fuchs et al., 2017a). The $k_{OH}$ instrument itself was located in the laboratory below the roof of the building. Therefore it can be assumed that the determined $k_{OH}$ values relate to a room temperature of 20°C, not to ambient temperature.

Sensitivity studies taking either ambient temperature or room temperature for the calculation of OH reactivity from measured OH reactant concentrations (see below) indicate that the effect of temperature differences on reaction rate



constants resulted in changes in the OH reactivity of typically less than 1% (maximum values 5 %) for conditions of this campaign.

The HONO data set used in the following model calculations is the average of the data from the FZJ-LOPAP and PKU-LOPAP instruments. In general, the measurements from the two LOPAP instruments showed good agreement

in the diurnal variation with a correlation coefficient of 0.96. However, the regression slope between FZJ-LOPAP and PKU-LOPAP was 0.618. A cross calibration by exchanging calibration standards during the campaign showed good agreement between the two instruments. The reason for the measurement discrepancy for ambient air remains unclear. The discrepancy between the two instruments is therefore considered as the uncertainty of the HONO measurements (± 20%, Table 1) which is larger than the usual calibration error (10%).

NO was measured by a commercial instrument using the chemiluminescence technique (Thermo Electron model 42i $NO$-$NO_2$-$NO_x$ analyzer). For $NO_2$, a custom-built photolytic-converter with a strong UV-LED at 395nm (residence time 1 s, temperature controlled sample air to 35°C, quartz surfaces when in contact with the illuminated sample air, conversion efficiency >95%) was used to convert $NO_2$. Ozone was measured by UV absorption (Thermo Electron model 49i). CO, $CO_2$, $CH_4$ and water vapor content were measured by a commercial instrument using the infrared

cavity ring-down technique (Picarro model G2401). $SO_2$ measurements were performed by a commercial instrument using pulsed UV fluorescence (Thermo Electron models 43i-TLE).

VOC measurements were performed by a gas chromatograph (GC) equipped with a mass spectrometer (MS) and a flame ionization detector (FID). The air sample was drawn into two parallel channels using a custom-built cooling device for enrichment before analysis that did not use liquid nitrogen but rather a sophisticated Peltier- chiller

(Wang et al., 2014). The GC-MS/FID instrument could provide measurements of $C_2$-$C_{11}$ alkanes, $C_2$-$C_6$ alkenes, and $C_6$-$C_{10}$ aromatics. In addition, formaldehyde (HCHO) was measured by a Hantzsch instrument (Aerolaser GmbH model AL4021). Other oxygenated VOCs were measured by a Proton Transfer Reaction - Time of Flight - Mass Spectrometer (PTR-ToF-MS), including methyl vinyl ketone and methacrolein, acetaldehyde, and acetone.

The surface and mass concentrations of aerosols were also measured, as well as the chemical composition and optical

properties, which will be presented and discussed in separate publications.

### 2.3 The 0-D model

A box model based on the Regional Atmospheric Chemical Mechanism version 2 (RACM2) updated with the newly proposed isoprene mechanism (Goliff et al., 2013; Peeters et al., 2014) was used to simulate the concentrations of the short-lived OH, $HO_2$ and $RO_2$ radicals. Measurements of photolysis frequencies, long-lived trace gases (NO, $NO_2$,

$O_3$, HONO, CO, $CH_4$, $C_2$−$C_{12}$ VOCs), and meteorological parameters were used to constrain the model by observations. Secondary species were not constrained but generated numerically by the model. A constant artificial loss rate that corresponds to an atmospheric lifetime of 24 hours was used for all unconstrained species to represent the effect of unspecified physical losses such as deposition, convection, and horizontal transportation. Sensitivity tests showed that the response of the calculated $RO_x$ concentrations to changes of this artificial loss rate by a factor of

two was less than 5%. The validity of the approach was tested by comparing the model results to parameters which were measured during the campaign but was not used as a model constraint. The comparison for oxygenated VOCs (e.g. acetaldehyde) and PAN showed agreement within 20% and 24%, respectively. Furthermore, $k_{OH}$ and



formaldehyde gave agreement within 10%, indicating that the model approach was able to describe the long-lived oxidation products reasonably well.

## 3 Results

### 3.1 Chemical and meteorological conditions

The observed trace gas concentrations were highly variable depending on the particular meteorological conditions. In Fig. 1, the HYSPLIT back trajectory analysis (Stein et al., 2015) shows two typical air mass transportation pathways. The dominating wind direction in Beijing during wintertime is Northwest from the Siberian tundra or from the Mongolian desert. The back trajectory analysis shows that the air parcels arriving in the city of Beijing either came directly from these remote regions bringing clean air to the UCAS site or first reach the Beijing downtown region

and then turned to the UCAS site, loaded with pollutants from the Beijing city region. As presented in Fig. 3, the CO concentration, a proxy for anthropogenic pollution, shows a distinct variation from day to day. The chemical conditions could be classified into three groups, namely background, clean, and polluted, respectively based on observed $k_{OH}$, CO, and $PM_{2.5}$. The thresholds polluted and the other two episodes are the daily average values of $k_{OH}$ (15 s$^{-1}$), which could be found in CO and $PM_{2.5}$ since they were highly correlated. The difference between the

background and clean episodes are whether there is a diurnal variation in the CO and $k_{OH}$, and $PM_{2.5}$. Only three days (22 and 23 January and 23 February) were classified as background conditions, during which strong northern winds (wind speed up to 10 m/s) were observed. The strong wind enhanced the dilution of pollutants coming from the Beijing region and prevented the build-up of a stable nocturnal surface layer. As shown in Fig. 4, the mean diurnal profiles of CO, $NO_2$, HONO, and $O_3$ were flat over most of the day. For the majority of the campaign, the

wind velocity was relatively small as reflected by the local wind speed measurements (less than 2 m/s). Hence, although the air masses originated over the Mongolian desert or the Siberian tundra, the observations at the UCAS site were slightly influenced by local emission from the nearby village and from close-by car traffic. The time periods for these relatively clean episodes are listed in Table 2.

        The pollution episodes happened on January 15, 16, 20, 21, 27, 28, 29, February 21 and March 1 to 4. The back

trajectory analysis shows that for those days the air masses were at least partly influenced by the Beijing city area which is located south of the UCAS site. In addition, the relative humidity increased from 28% for the clean episodes to 44% for the pollution episodes for daytime averaged conditions, suggesting that the site was influenced by humidified air from the south. The concentrations of trace gases and particle matters increased noticeably during the pollution episodes (Table 2).

The dissimilarities between different episodes can be easily illustrated using the mean diurnal profiles (Fig. 4). Solar radiation, indicated by the measured photolysis frequencies e.g. j(O$^1$D), was comparable during the background and the clean episodes but reduced by about 20~30% during the pollution episodes (Table 2). NO concentrations were below 1 ppbv during the background episodes. The diurnal peak of NO (4ppbv) appeared at 09:00 during the clean episodes. During the polluted episodes, NO concentrations increased significantly to reach a maximum of 15 ppbv

around sunrise. $O_3$ remained nearly constant at 38 ppbv at all times during daytime of the background episodes, which can be regarded as a continental wintertime $O_3$ background concentration. During clean episodes, the wind



velocity was reduced, so that local NO emissions could accumulate and titrate away the ozone, especially during nighttime when vertical mixing was reduced. $NO_2$ showed an anti-correlation with $O_3$ because of this inter-conversion. Besides the titration effect, there were deposition processes and non-photochemical reactions which additionally diminished the concentration of ozone (and of $NO_2$) in the shallow surface layer during night hours.

However, the maximum $O_3$ concentrations during afternoon hours, when vertical mixing inhibited the accumulation of the local NO emissions, were comparable to those observed during the background episodes (Fig. 4). These processes led to a slow reduction of the near-surface ozone concentration starting at sunset, to a minimum just at sunrise, and to a fast recovery within the next 2-3 hours. This increase of $O_3$ in the morning hours during the clean episodes was at least partly a result of entrainment of background $O_3$ from the residual layer.

During the pollution episodes, the ozone concentrations showed a distinct diurnal variation due to a strong titration by NO. The concentration of $NO_2$ was high at night but decreased after sunrise, which could be due to the poor dilution conditions at night and fast photolysis after sunrise. $O_x$ is the sum of $NO_2$ and $O_3$, which is considered as a more conservative metric than $O_3$ because it is not affected by the interruption of fresh NO emission, especially in urban environments (Kley et al., 1994; Kleinman et al., 2002). $O_x$ increased from 30 ppbv at 07:00 to 50 ppbv at

16:00 during the pollution episodes. After sunset, $O_x$ started to decrease due to physical losses, like deposition and transportation of $O_3$ and $NO_2$, or chemical conversion to $N_2O_5$. The fast increase in $O_x$ concentrations during daytime was a specific feature for the pollution episodes, which indicated the strong photochemistry happening during the haze events.

$SO_2$ is a tracer for regional air pollution originating from coal combustion, e.g. power plants and residential heating.

For background and clean episodes, $SO_2$ exhibited low concentrations around 1 ppbv. It increased from 5 ppbv at sunrise to 10 ppbv at 16:00 during the pollution episodes. The large increase of $SO_2$ concentrations during pollution episodes which showed a distinct diel increase could be related to the pollutants accumulation process.

Typically, HONO accumulated during the night (most probably from heterogeneous reactions of $NO_2$ on humid surfaces) and started to decrease after sunrise due to fast photolysis. This was observed during clean or polluted

episodes when the measured HONO showed a distinct diurnal minimum during the afternoon and started to increase between sunset and sunrise up to values around 1 ppbv during the night. In contrast, the averaged HONO concentrations were about 0.05 ppbv without obvious diurnal variation for background conditions (Table 2).

### 3.2 Comparison between measured and calculated OH, $HO_2$, $RO_2$ radical concentrations and OH reactivity

Fig. 5 shows the measured and modeled OH, $HO_2$, $RO_2$ radical concentrations and OH reactivity. The largest OH

concentration appeared around noontime with large day-to-day variability. The daily maximum varied in the range $1 \times 10^6$ cm$^{-3}$ to $1 \times 10^7$ cm$^{-3}$. The highest OH concentrations were observed during the second half of the campaign (27 and 28 February) when the temperature was above the freezing point and the photolysis frequencies for the short wavelength region, e.g. $j(O^1D)$, were higher than in January (Fig. 3).

In general, the model could reproduce the observed OH concentrations within 30%. It is important to notice that the

good agreement between measured and modeled OH concentrations was mainly due to the availability of observed HONO concentrations. A sensitivity test showed that the calculated OH radical concentrations were reduced by 43% if the model was not constrained to the HONO measurements. This underlines that the currently known gas-phase



formation of HONO from the reaction of OH with NO is not sufficient to sustain the high HONO concentrations observed during this winter campaign. The result agrees with numerous field and laboratory studies which have reported additional daytime sources of HONO (e.g., Kleffmann et al., 2005), for example from the heterogeneous conversion of nitrogen species (nitrates, nitric acid, $NO_x$) at surfaces and soil emissions (see overview in Meusel et

al. 2018, and references therein). The present study demonstrates that at least some of the additional HONO sources are also effective at cold conditions and play an important role for the winter-time oxidation of pollutants in urban atmospheres. It also confirms the need for in-situ HONO measurements during field studies focusing on the investigation of atmospheric radical chemistry (Alicke et al., 2003; Su et al., 2011; Lu et al., 2012; Kim et al., 2014).

The overall correlation between observed OH concentrations and photolysis frequencies is shown in Fig. 6. Both

observed and modeled OH concentrations show a good correlation with $j(O^1D)$ and with $j(NO_2)$ with the coefficient of determination $R^2$ being larger than 0.7, which can be expected in summer when the radical chemistry is initiated mainly by photolysis processes including ozone photolysis. However, $j(O^1D)$ increased by a factor of 2 from the start to the end of the campaign while $j(NO_2)$ remained nearly constant (Fig. 3). The correlation between calculated OH and $j(NO_2)$ is more compact and linear than that between OH and $j(O^1D)$, reflected in a slightly higher correlation

coefficient (Fig. 6). In the model, the OH concentrations tend to depend strongly on $j(NO_2)$. This can be explained by HONO photolysis being the dominant primary source of OH radicals (see section 4.1) and the well-known linear correlation between $j(NO_2)$ and $j(HONO)$ (Kraus and Hofzumahaus, 1998). The OH observations do not show such a difference, $j(O^1D)$ and $j(NO_2)$ both exhibit the same good correlation coefficient.

Despite the general agreement, the model systematically underestimates OH concentrations for two days (27 and 28

February) by a factor two (Fig. 5). The OH reactivity was relatively small on these days (<10 $s^{-1}$). A model sensitivity run with an additional primary OH source of 0.25 ppbv/h shows that the modeled OH concentrations would increase by 50%, which would be enough to close the gap between measured and modeled OH for these two days. However, the possibility of an instrument failure of the PKU-LIF during these two days cannot be excluded. An indication may be that the data points on these two days do not follow the generally very good OH-JO1D

correlation (Fig. 6). Therefore, we excluded these two days from the analysis.

The daytime averaged peroxy radical concentrations are smaller by a factor of 5 compared to summertime at the rural site of Wangdu in the North China Plain (Tan et al., 2017). The strong reduction in radical concentrations is due to the attenuated solar radiation and thus smaller primary radical sources. In addition, $NO_x$ concentrations during the BEST-ONE campaign were larger resulting in a faster peroxy radical loss and thus shorter lifetimes.

For the background episodes, the calculated $RO_2$ data are not shown in Fig. 7 because the observed NO concentrations were close to the limit of detection of the NO instrument and thus the model was highly sensitive to the fluctuations of the NO measurements around zero (Fig. 5). The observed $RO_2$ radical concentrations do not show a significant difference between background, clean, and pollution episodes (Fig. 7) while a large difference can be observed for the calculated $RO_2$ concentrations. Both the calculated OH and $HO_2$ radical concentrations were lower

than observed during polluted days. Overall, during clean days, the model can reproduce the observed OH, $HO_2$ and $RO_2$ radical concentrations with an observed-to-modeled ratio of 1.31, 1.03 and 0.98 for daytime conditions, respectively (Fig. 5). The nighttime $RO_2$ radicals are often overestimated by the model calculations by up to a factor of ten, which could be the result of a NO measurement artifact. The observed NO concentrations were often below





the limit of detection of the $NO_x$ instrument (60pptv). As a result, a small bias in the NO measurement could lead to an unrealistic long lifetime of $RO_2$ radicals in the model. A model sensitivity run showed that $RO_2$ concentrations could be significantly reduced if the constrained NO concentrations in the model were increased by 20 pptv, which is within the precision of the instrument. A loss of NO inside the inlet-line of the instrument would already be enough

to explain this effect that was within the limit of detection.

The model underestimates the observed OH concentrations by a factor of 1.8 during polluted days, and the peroxy radical concentrations are significantly underestimated by up to a factor of 5 (Fig. 7). A detailed analysis of these days is given in section 4.2. A chemical model based on MCM 3.3.1 also predicts similar results as RACM2, suggesting that such underestimation is not a result of the "family approach" for organic molecules used in RACM2.

The model was capable to reproduce the directly observed OH reactivity within 10% during all episodes by including the reactivity of observed VOCs (about 70% of the observed $k_{OH}$) and the estimated contributions from OVOCs calculated by the model (Fig. 7). The speciation of the total OH reactivity showed that the major OH reactants were $NO_x$ and CO. For the polluted episodes, the average OH reactivity increased from 10 to 26 s$^{-1}$ with a significant increase in the relative contributions from the inorganic compounds. On average, CO and $NO_x$ contributed 23% and

37% to the total OH reactivity, respectively.

## 4. Discussion

### 4.1 Sources and sinks of $RO_x$ radicals

As shown in Fig. 8, the radical chain reactions were mainly initiated by photolysis processes. HONO photolysis was the most important radical primary source. It contributed up to 46% (averaged rate 0.26 ppbv/h) of the total primary

production rate of radicals for daytime conditions. During the BEST-ONE campaign, the production rates of radicals varied slightly between different episodes (Fig. 9). The production rates were highest during the polluted episodes compared to the background and clean episodes. The major difference was due to the contribution from HONO photolysis, which accounted for the major source in all cases contributing 25% (0.10 ppbv/h), 40% (0.21 ppbv/h), and 55% (0.37 ppbv/h) of diurnally averaged rates of 0.10, 0.21, 0.37 ppb/h for background, clean and polluted

episodes, respectively. Alkene ozonolysis was the major radical source of nighttime and the second largest primary radical source during the day (28%, 0.16 ppbv/h). The relative importance of the ozonolysis of alkenes was slightly larger during the background (39%) and clean (33%) episodes. The ozone photolysis was almost negligible for our wintertime BEST-ONE campaign. At high solar zenith angles, short wavelength radiation (<320 nm) is suppressed due to the strong attenuation by the longer pathway through the ozone layer in the stratosphere. In addition, the water

vapor mixing ratio at the low winter temperatures decreases to below 0.3%, one order of magnitude lower than during summertime. Formaldehyde photolysis contributed on average 17% to the total $RO_x$ primary production. The photolysis of carbonyls compounds calculated by the model also made a noticeable portion of the total primary production of radicals, about 7% (Fig. 8).

The importance of HONO photolysis is reported in previous studies for other locations, both urban and suburban

(Alicke et al., 2003; Dusanter et al., 2009; Mao et al., 2010; Ren et al., 2013). For winter campaigns, HONO photolysis was found to be the dominant (>50%) daytime $RO_x$ radical source in New York City, NYC (Ren et al.,





2006), Tokyo (Kanaya et al., 2007) and in Boulder in Colorado (Kim et al., 2014). During the PUMA campaign in Birmingham, HONO photolysis contributed only 36% to the total OH radical primary sources (Emmerson et al., 2005b). While the reported value of 36% is a lower limit, since HONO was not measured. Instead, it was calculated from the equilibrium between HONO and OH+NO. As OH and NO can recombine to nitrous acid, the net effect of

HONO photolysis to radical production might be partly compensated. Kanaya et al. (2007) found that the OH+NO (+M) reaction was balanced by the HONO photolysis during the morning hours. In our study, the gas-phase HONO formation only compensated 10% of the observed HONO photolysis. Therefore, the net effect of HONO photolysis remains the dominant radical source in wintertime during the BEST-ONE campaign. In Fig. 9, the radical production for the Wangdu summer campaign is shown in comparison to the current campaign. The total daytime radical

production rate was reduced in winter by a factor of 6 compared to summertime. The contribution of HONO photolysis also changed, contributing 39% to the total radical production for the Wangdu summer campaign.

The importance of alkene ozonolysis to the primary $RO_x$ radical production was also found in the IMPACT winter campaign in Tokyo, where it contributed 49% of the $RO_x$ production on a 24 h basis (Kanaya et al., 2007). In other campaigns, alkene ozonolysis was comparable to HONO photolysis even during daytime, which contributed 42% in

NYC (Ren et al., 2006) and 63% in Birmingham (Emmerson et al., 2005b), respectively. During the summertime in Wangdu, the contribution from alkene ozonolysis was reduced to 15% but the absolute rate was larger (0.47 ppbv/h) than in Huairou/Beijing wintertime (0.16 ppbv/h).

Different from our study, the photolysis of ozone was found to be relatively important in a rural site of Colorado, contributing 15% to the OH primary production for noontime conditions. The absolute radical production rate is

comparable in these two campaigns (Table 3), the higher contribution of ozone photolysis is due to the fact that the Colorado campaign took place at a slightly later time (February and March) of the year (Kim et al., 2014).

The photolysis of carbonyl compounds can be an important radical source for urban and suburban areas (Emmerson et al., 2005b). HCHO photolysis contributed up to 6% of the total $HO_x$ (=OH+HO$_2$) production in NYC (Ren et al., 2006) and 10% in Tokyo (Kanaya et al., 2007) for wintertime conditions, comparable to their contributions in

summertime, for example 8% for NYC (Ren et al., 2006) and 18% for Tokyo (Kanaya et al., 2007).

The termination of $RO_x$ radicals is achieved either by the reaction with $NO_x$ or by the peroxy radical self-reactions. The $NO_x$ termination reactions can be further subdivided into OH+NO, OH+NO$_2$, RO$_2$+NO, and RO$_2$+NO$_2$. As shown in Fig. 8, the reaction between OH and NO$_2$ dominated the total radical termination process during the BEST-ONE campaign, which contributed 49% of the total $RO_x$ loss for daytime average. The equilibrium between peroxy

radicals and PAN-type compounds was a sink for $RO_x$ radicals due to the low ambient temperature during the winter campaign. The net PAN-type compounds species formation had a noticeable impact on the radical budget (25% daytime averaged). Since the observed and modeled PAN concentrations agree within 24%, it gave confidence to the net PAN formation rate. In London, the net PAN formation was also found to be a major radical sink, contributing 35% to the total radical destruction rate (Whalley et al., 2018). The $RO_x$ self-reactions were nearly negligible (<3%) as the

peroxy radical concentrations were small (<$1\times10^8$ cm$^{-3}$). For comparison, this is one order of magnitude smaller than in the summertime Wangdu campaign, where the radical termination process was mainly dominated by the hydroperoxide formation path (Fig. 9).



The relative importance of various loss pathways was compared between the different episodes (Fig. 9). The reaction between OH and $NO_2$ contributed 40% and 61% to the total $RO_x$ loss during clean and polluted episodes. The net loss of PAN-type compounds became of relative importance during background episodes, contributing 50% to the total radical termination because of the lower temperatures (Table 2).

### 4.2 Model-measurement comparison of peroxy radicals concentrations

As shown in Fig. 10, the observed $RO_2$ concentrations were relatively constant over the whole NO regime. However, the model predicted a strong decreasing trend with higher NO concentrations. This is further illustrated by the observed-to-modeled ratio, which increased from a value of 1 at 1 ppbv of NO to 9 at 6 ppbv of NO (Fig. 11). As shown in Fig. 11, the underestimation of both $HO_2$ and $RO_2$ radical concentrations became larger with higher $NO_x$. In contrast, the observed-to-modeled ratio of OH concentrations was almost constant with a value of 1.5 over the full $NO_x$ regime, which is within the combined uncertainties of measurement and model calculations. An underestimation of $HO_2$ radical concentrations by the model at high $NO_x$ values was also observed in previous studies (Shirley et al., 2006; Ren et al., 2006; Emmerson et al., 2007; Kanaya et al., 2007; Ren et al., 2013; Griffith et al., 2016). It was suggested that one reason for this discrepancy might be due to the poor mixing and segregation between NO and peroxy radicals (Dusanter et al., 2009). Alternatively, it can also imply missing peroxy radical sources in the current chemical mechanisms relevant in particular for the high $NO_x$ regime. In addition, the underestimation of the measured $RO_2$ radicals by the model could explain partly the discrepancy observed for $HO_2$ radicals due to insufficient recycling from the OH oxidation chain.

In the present study, the radical cycling between OH and the peroxy radicals is relatively well constrained due to the availability of measured $k_{OH}$ and NO. Since the model reproduces $k_{OH}$ within 10%, unmeasured VOCs cannot be responsible for a possible underestimation of the peroxy radical production resulting from the reaction of VOCs with OH. As the NO concentration in the model was constrained to measurements, also the main loss processes of peroxy radicals in the high NOx regime, the reactions with NO, are well determined. In addition, a possible segregation effect by the fast variability of radical precursors in combination with non-synchronous observations was minimized by the experimental setup. The inlets of the LIF instrument and of other instruments were very close to each other (within 2 m). The sampling height was about 20 m above ground, 50 m away from the next street. Therefore, segregation is expected to play only a minor role. This is supported by the observed $O_3$, NO, and $NO_2$ concentrations, which were close (within 10%) to a steady state. All these arguments indicate that the significant peroxy radical underestimation is probably caused by missing primary sources in the model.

Based on the discussion above, we can assume that the peroxy radicals were in the steady state so that their production and destruction rates were balanced at all times. The production of peroxy radicals $P(HO_2+RO_2)$ can be calculated as follows:

$$P(HO_2+RO_2) = k_{VOC}\times[OH] + P(HO_2)_{primary} + P(RO_2)_{primary} \tag{3}$$

Here, $k_{VOC}$ denotes the part of OH reactivity which is caused by CO, $CH_4$, VOC, and OVOCs reactions. This value is equal to the difference between observed total OH reactivity and $NO_x$ reactivity. The known $HO_2$ and $RO_2$ primary sources contribute less than 5% compared to the radical recycling rate at high $NO_x$ conditions.

The loss rate of $HO_2$ and $RO_2$ can be expressed as:



$$L(HO_2+RO_2) = k_{HO2+NO}\times[HO_2]\times[NO] + L(HO_2)_{termination} + L(RO_2)_{termination} \qquad (4)$$

For high $NO_x$ conditions, the termination rates for $HO_2+HO_2$, $HO_2+RO_2$, $RO_2+RO_2$ are less than 1% compared to the reaction rates between $HO_2$ and NO. We also tested the equilibrium between $HO_2$ and $HNO_4$ as well as between $RO_2$ and PAN-type compounds. They had only a minor impact on the $RO_x$ budget calculation (<4%). The reaction of $RO_2$ and NO is not in equation (4), because this reaction converts to $HO_2$ and thus is not an effective loss of peroxy radicals.

Because the production and destruction rate of peroxy radicals must be balanced, the missing peroxy radical source $P'(RO_x)$ can be determined by the difference between known radical loss rate and production rate. It is worth noting that all primary sources and termination processes of $HO_2$ and $RO_2$ are negligible compared to the radical propagation.

$$P'(RO_x) = k_{HO2+NO}\times[HO_2]\times[NO] + P(HO_2)_{primary} + P(RO_2)_{primary} - k_{VOC}\times[OH] - L(HO_2)_{termination} - L(RO_2)_{termination} \qquad (5)$$

As shown in Fig. 12, $P'(RO_x)$ is essentially zero in background air. In the clean and polluted case, however, a significant missing radical source is found during daytime. On polluted days, $P'(RO_x)$ is very large and shows a broad maximum with a peak value of about 5 ppbv/h at 10:00. The production rate is 2.5ppbv/h on average, nearly a factor of 5 larger than the known primary source of $RO_x$ during the polluted days (0.62 ppbv/h). The error of the $P'(RO_x)$ determination is estimated to be ±1.7 ppbv by considering the uncertainty from measurement and kinetic reaction rates. On the clean days (Fig. 12), the missing radical source is significant only during morning hours, peaking at 3 ppbv/h at 10:00.

A missing primary radical source was already proposed to explain a morning $RO_2$ underestimation by the model during the summer campaign in Wangdu (Tan et al., 2017). The proposed source originated from photolysis of $ClNO_2$ generating chlorine radicals, which can oxidize VOC to $RO_2$. However, the observed $ClNO_2$ (0.5 ppbv, maximum of diurnal average, Tham et al., 2016) and $Cl_2$ (Liu et al., 2017) concentrations could only explain 10% and 30% of the missing primary $RO_x$ source for the summer Wangdu campaign (Tan et al., 2017). During the winter campaign in Beijing, the missing primary $RO_x$ source was 2 to 3 times larger than what was found during the Wangdu summer campaign. Since $ClNO_2$ and molecular chlorine were not measured at Huairou, their possible role in the production of $RO_x$ cannot be quantified here. Implementing a generic primary $RO_2$ radical source in the model allows increasing the modeled $HO_2$ and $RO_2$ concentrations. However, matching the modeled $RO_2$ concentrations to the observations leads to an overprediction of OH by more than a factor of 5. To maintain a relatively good agreement between observed and modeled OH, an additional OH sink would be needed, but this is difficult to reconcile with the good agreement between the measured and modeled $k_{OH}$.

From the time series (Fig. 3) and the mean diurnal profiles (Fig. 4) of $O_x$, it is possible to deduce that excess $O_x$ higher than the typical $O_x$ value of 40 ppbv during afternoon only appeared during the pollution days. This indicates that the $O_x$ was produced by local photochemical reactions. The maximum instantaneous ozone production rate (not including any $O_x$ potential loss process) can be approximated by the oxidation rate of NO by $HO_2$ and $RO_2$:

$$P(O_3) = k_{HO2+NO}[HO_2][NO] + \sum k_{RO2,i+NO}[RO_2]_i[NO] \qquad (6)$$

The base model underpredicts the observed $HO_2$ and $RO_2$ concentrations significantly leading to the strong underestimation of this local ozone production rate (Fig. 12). As discussed above, an additional primary radical source is required to close the $RO_x$ budget, which would be the dominant contributor to the fast ozone production



rate during pollution episodes. Similarly, ozone production rates that were directly measured in cities in the US were significantly higher than model calculations for high $NO_x$ regimes (Baier et al., 2017; Brune et al., 2016; Cazorla et al., 2012).

### 4.3 High OH concentration in Beijing during wintertime

The observed OH concentrations presented in this study are nearly one order of magnitude larger than global models predict for northern China in winter (Lelieveld et al., 2016; Huang et al., 2014). The higher-than-expected OH concentrations indicate that the oxidation capacity of the atmosphere was high in Beijing and in the North China Plain in wintertime. As shown in Table 3, the averaged OH concentrations in Beijing for January were also higher than at other mid-latitude field studies in which OH concentrations were measured in winter. In this study, the

observed OH radical concentrations at noontime ranged from $2.4 \times 10^6$ cm$^{-3}$ in severely polluted air ($k_{OH} \sim 27$ s$^{-1}$) to $3.6 \times 10^6$ cm$^{-3}$ in relatively clean air ($k_{OH} \sim 5$ s$^{-1}$). The reported OH maximum concentrations for urban and suburban locations in the northern hemispheric winter were $1.4 \times 10^6$ cm$^{-3}$ in NYC (Ren et al., 2006), $1.7 \times 10^6$ cm$^{-3}$ in Birmingham (Heard et al., 2004) and $1.5 \times 10^6$ cm$^{-3}$ in Tokyo (Kanaya et al., 2007), respectively (Table 3). Comparably high OH concentrations of up to $2.7 \times 10^6$ cm$^{-3}$ were observed in a rural site in Colorado in late February

(Kim et al., 2014). In our study, the OH maximum reached $5 \times 10^6$ cm$^{-3}$ from February 20 to 28 with similar j(O$^1$D) values (maximum $1 \times 10^{-5}$ s$^{-1}$ on average) as observed in the Colorado site. On the other hand, the OH reactivity was on average less than 5 s$^{-1}$, which resulted in much slower OH turnover rates at that rural site in the US with an estimated maximum of 1.6 ppbv/h (Kim et al., 2014). The combination of high OH concentration and moderate OH reactivity at Huairou/Beijing resulted in a fast oxidation rate by OH up to 3.6 ppbv/h for the campaign average

conditions.

The primary radical production rate at Huairou/Beijing was in the lower range among all other winter $HO_x$ campaigns (Table 3). However, the high observed OH concentrations and moderate OH reactivities led to a fast OH turnover rate, which had to be sustained by a large radical chain length. The radical chain length is used to describe the efficiency of radical propagation. It can be calculated as following.

$$ChL = [OH] * k_{VOC} / P(RO_x) \qquad (7)$$

The radical chain length was up to 5 in Huairou/Beijing, but was observed to be in the range from 2 to 3 in other winter campaigns (Table 3). This indicates radical propagation is very efficient during the BEST-ONE campaign. However, if a radical primary production is missing in the calculation, the calculated radical chain length would be too large. Therefore, for this study, the chain length should be regarded as an upper limit. For example, in the

NACHTT campaign, only the OH production rate was taken into account by Kim et al. (2014), which leads to an overestimation of the radical chain length.

Secondary products are believed to be major contributors to aerosol particles in haze events (Guo et al., 2014; Cheng et al., 2016; Wang et al., 2016; Wang et al., 2017). An important question is what is the contribution of gas-phase reactions to haze formation. Previous studies attempted to quantify the contributions of gas-phase and aerosol-phase

reactions to oxidation processes during wintertime by chemical models (Huang et al., 2014; Guo et al., 2014; Nan et al., 2017). In this study, the in-situ observations of OH radical provide an experimental constraint to quantify the sulfate and nitrate production by OH oxidation. As discussed above, the high OH concentrations indicate a fast gas





phase oxidation during wintertime in Beijing. As a consequence, the nitrate acid production rate P(HNO$_3$) during wintertime at Huairou/Beijing was 0.28 ppbv/h for daytime averaged, only 3 times slower compared to the campaign in Wangdu during summertime (0.81 ppbv/h). In comparison, the SO$_2$ oxidation rate was 0.02 ppbv/h. One should keep in mind that the formation of gas phase sulfate and nitrate via OH oxidation does not necessarily lead to particle

formation, due to the dependence on the gas-aerosol partitioning. The relative fast oxidation rate presented here only shows the particle growth potential from gas phase reactions. Nevertheless, the in-situ measurements presented in this study provide experimental evidence for a strong gas-phase production potential of aerosol precursors, which underlines the importance of taking OH chemistry into account for the understanding of the wintertime haze formation.

**5. Summary and Conclusions**

The BEST-ONE campaign was performed at Huairou at the northwest edge of Beijing in 2016 to elucidate pollution formation mechanisms in the North China Plain providing the first wintertime observations of OH, HO$_2$ and RO$_2$ radicals in this region. Relatively high radical concentrations were observed compared to other winter campaigns in suburban and urban environments at similar latitudes. OH radical concentrations at noontime ranged from 2.4×10$^6$

cm$^{-3}$ in severely polluted air ($k_{OH}$ ~ 27 s$^{-1}$) to 3.6×10$^6$ cm$^{-3}$ in relatively clean air ($k_{OH}$ ~ 5 s$^{-1}$), two-fold higher than those observed in Birmingham (Heard et al., 2004), Tokyo (Kanaya et al., 2007), and NYC (Ren et al., 2006) during wintertime. OH reactivity was measured simultaneously during this campaign and showed a large variability in values between 5 s$^{-1}$ and 95 s$^{-1}$. The experimentally determined OH cumulative turnover rates were on average 20.7 ppbv per day, indicating a fast gas-phase oxidation capacity in this region even in wintertime. During a strong haze

event in early March at the end of the campaign, the OH turnover rate increased to more than 20 ppbv/h, which is comparable to summertime conditions.

In addition to the radical observations, numerous parameters were measured during the BEST-ONE campaign revealing the key processes of radical chemistry for wintertime pollution episodes. The photolysis of O$_3$, the globally most important source of OH radicals, was strongly reduced due to the low UV-B levels and the small concentration

of water vapor during wintertime. The initiation of RO$_x$ radical chain reactions was instead dominated by the photolysis of HONO, which contributed about 46% of the observed total primary radical production rate. Most of the observed HONO cannot be explained by the gas-phase production from the OH+NO reaction. Similar to summertime, other HONO sources must be dominating the HONO production thereby playing a critical role in the photochemical oxidation of pollutants in urban air in wintertime. The alkene ozonolysis reactions were the second

important radical source, contributing 28% to the total primary source. The radical chain termination processes were dominated by the reaction between OH and NO$_2$ (49%).

The comparison of radical concentrations between observations and box model simulations based on long-lived trace gas measurements showed only 30% difference during clean air episodes. However, RO$_2$ radical concentrations were underestimated by a factor of 5 by the model in the high NO$_x$ regime. This severe underestimation of RO$_2$ suggests

that an important radical source is missing in the current chemical mechanism during the pollution episodes. As a consequence, the production of secondary pollutants like ozone driven by this missing radical source during wintertime haze events in Beijing could be strongly underestimated.





**Acknowledgment.**

We thank the science teams of the BEST-ONE campaign. This work was supported by the National Natural Science Foundation of China (Major Program: 21190052 and Innovative Research Group: 41121004, Grant No. 21522701, 41375124), the National Science and Technology Support Program of China (No.2014BAC21B01), the Strategic

Priority Research Program of the Chinese Academy of Sciences (grant no. XDB05010500), the Collaborative Innovation Center for Regional Environmental Quality, the EU-project AMIS (Fate and Impact of Atmospheric Pollutants, PIRSES-GA-2011-295132) and the BMBF project: ID-CLAR (01DO17036). The authors gratefully acknowledge the NOAA Air Resources Laboratory (ARL) for the provision of the HYSPLIT transport and dispersion model and READY website (http://www.ready.noaa.gov) used in this publication.

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



**Table 1 Measured species and performance of the instruments**

| Parameters | Measurement technique | Time resolution | Detection limit [a] | Accuracy |
|---|---|---|---|---|
| OH | LIF [b] | 30 s | $4.0\times10^5 cm^{-3}$ | ±10% |
| HO$_2$ | LIF [b, c] | 30 s | $1.0\times10^7 cm^{-3}$ | ±13% |
| RO$_2$ | LIF [b, c] | 30 s | $0.5\times10^7 cm^{-3}$ | ±11% |
| $k_{OH}$ | LP-LIF [d] | 90 s | $0.3 s^{-1}$ | ±10% ±0.7 s$^{-1}$ |
| Photolysis frequencies | spectroradiometer | 20 s | [e] | ±10% |
| O3 | UV photometry | 60 s | 0.5 ppbv | ±5% |
| NO | chemilumincence | 60 s | 60 pptv | ±20% |
| NO$_2$ | chemilumincence [f] | 60 s | 300 pptv | ±20% |
| HONO | LOPAP [g] | 30 s | 7 pptv | ±20% |
| CO,CH$_4$,CO$_2$,H$_2$O | CRDS | 60 s | [h] | [i] |
| SO$_2$ | Pulsed UV fluorescence | 60 s | 0.1 ppbv | ±5% |
| HCHO | Hantzsch fluorimetry | 60 s | 25 pptv | ±5% |
| Volatile organic compounds [j] | GC-FID/MS [k] | 1 h | (20~300) pptv | ±(15~20) % |
| Oxygenated organic compounds | PTR-ToF-MS | 10 s | (50-100) pptv | ±(10~15)% |

[a] signal to noise ratio = 1; [b] Laser-induced fluorescence; [c] Chemical conversion via NO reaction before detection; [d] Laser photolysis – laser-induced fluorescence; [e] Process-specific, 5 orders of magnitude lower than maximum at noon; [f] Photolytic conversion to NO before detection, home built converter; [g] Long-path absorption photometry; [h]

5 Species-specific, for CO: 1 ppbv; CH$_4$:1 ppbv; CO$_2$: 25 ppbv; H$_2$O: 0.1% (absolute water vapor content).; [i] Species-specific, for CO: ±1 ppbv; CH$_4$: ±1 ppbv; CO$_2$: ±25 ppbv; H$_2$O: ±5 %; [j] VOCs including C$_2$-C$_{11}$ alkanes, C$_2$-C$_6$ alkenes, C$_6$-C$_{10}$ aromatics; [k] Gas chromatography equipped with a mass spectrometer and a flame ionization detector.



**Table 2 Comparison of observed parameters for different episodes (24 h average values with 1σ standard deviations of ambient variabilities)**

| Parameter | Background | Clean | Polluted |
|---|---|---|---|
| Temperature / °C | -10.2 ± 6.3 | -4.0 ± 4.9 | -3.9 ± 4.4 |
| Pressure / hPa | 1025 ± 4 | 1015 ± 4 | 1012 ± 6 |
| RH / % | 19.6 ± 8.3 | 29.5 ± 11.7 | 46.3 ± 17.2 |
| Wind speed / m s$^{-1}$ | 3.7 ± 1.6 | 2.4 ± 1.3 | 1.9 ± 0.8 |
| j(O$^1$D) / 10$^{-5}$s$^{-1}$ | 0.67 ± 0.24 | 0.63 ± 0.23 | 0.56 ± 0.29 |
| j(NO$_2$) / 10$^{-3}$s$^{-1}$ | 6.52 ± 0.78 | 6.04 ± 0.63 | 4.23 ± 1.66 |
| OH / 10$^6$cm$^{-3}$ | 2.73 ± 1.19 | 3.58 ± 2.32 | 2.36 ± 0.74 |
| HO$_2$ / 10$^8$cm$^{-3}$ | 1.04 ± 0.62 | 0.93 ± 0.72 | 0.52 ± 0.23 |
| RO$_2$ / 10$^8$cm$^{-3}$ | 0.70 ± 0.34 | 0.76 ± 0.46 | 0.71 ± 0.41 |
| O$_3$ / ppb | 34.7 ± 4.7 | 27.9 ± 9.0 | 11.6 ± 10.1 |
| NO / ppb | 0.30 ± 1.61 | 2.06 ± 7.93 | 9.27 ± 12.54 |
| NO$_2$ / ppb | 2.0 ± 4.0 | 10.3 ± 13.6 | 32.5 ± 15.1 |
| HONO / ppb | 0.05 ± 0.05 | 0.23 ± 0.24 | 0.98 ± 0.90 |
| CO / ppm | 0.22 ± 0.22 | 0.33 ± 0.19 | 1.19 ± 0.70 |
| HCHO / ppb | 0.47 ± 0.38 | 1.29 ± 0.69 | 2.41 ± 0.97 |
| $k_{OH}$ / s$^{-1}$ | 5.4 ± 3.7 | 10.1 ± 5.6 | 26.9 ± 9.5 |
| PM$_{2.5}$ / µg/m$^3$ | 0 ± 20 | 6 ± 15 | 55 ± 36 |
| Dates | 01.22,01.23,02.23 | 01.06,01.07,01.10,01.11,01.03, 01.24,01.25,01.26,01.31,02.01, 02.20,02.24,02.25,02.26,02.27, 02.28,02.29 | 01.15,01.16,01.20, 01.21,01.27,01.28, 01.29,02.21,03.01, 03.02, 03.03, 03.04 |

j(O$^1$D)、j(NO$_2$)、OH、HO$_2$ and RO$_2$ concentrations are noontime averaged values () peak values of of 1-hour averages.



**Table 3 Noontime averaged concentrations of OH, NO$_2$ and OH reactivity and total radical production rate for campaigns investigating photochemistry including OH radical measurements during wintertime.**

| | OH 10$^6$ cm$^{-3}$ | P(RO$_x$) ppbv/h | $k_{OH}$ s$^{-1}$ | NO$_2$ ppbv | Chain length | reference |
|---|---|---|---|---|---|---|
| Birmingham, UK (2000, Jan-Feb) | 1.7 | 2.8 | 30[a] | 9.3 | 2.1 | Heard et al., (2004); Emmerson et al., (2005a, b) |
| NYC, US (2004, Jan-Feb) | 1.4 | 1.4 | 27 | 15 | 3.3 | Ren et al., 2006 |
| Tokyo, JP(2004, Jan-Feb) | 1.5 | 1.4 | 23[a] | 12 | 3.1 | Kanaya et al., (2007, 2008) |
| Boulder, US (2011, Feb-Mar) | 2.7 | >0.7[b] | 5[c] | 5 | <2.0 | Kim et al., (2014) |
| Beijing, CN (2016, Jan-Mar) | 2.8 | 0.9 | 12 | 6 | 4.7 | This study |

[a] $k_{OH}$ is calculated from the model
[b] only OH production rate is available, therefore should be regarded as the lower limit
5   [c] the sum of reactivity from VOC, OVOC, and NO$_2$





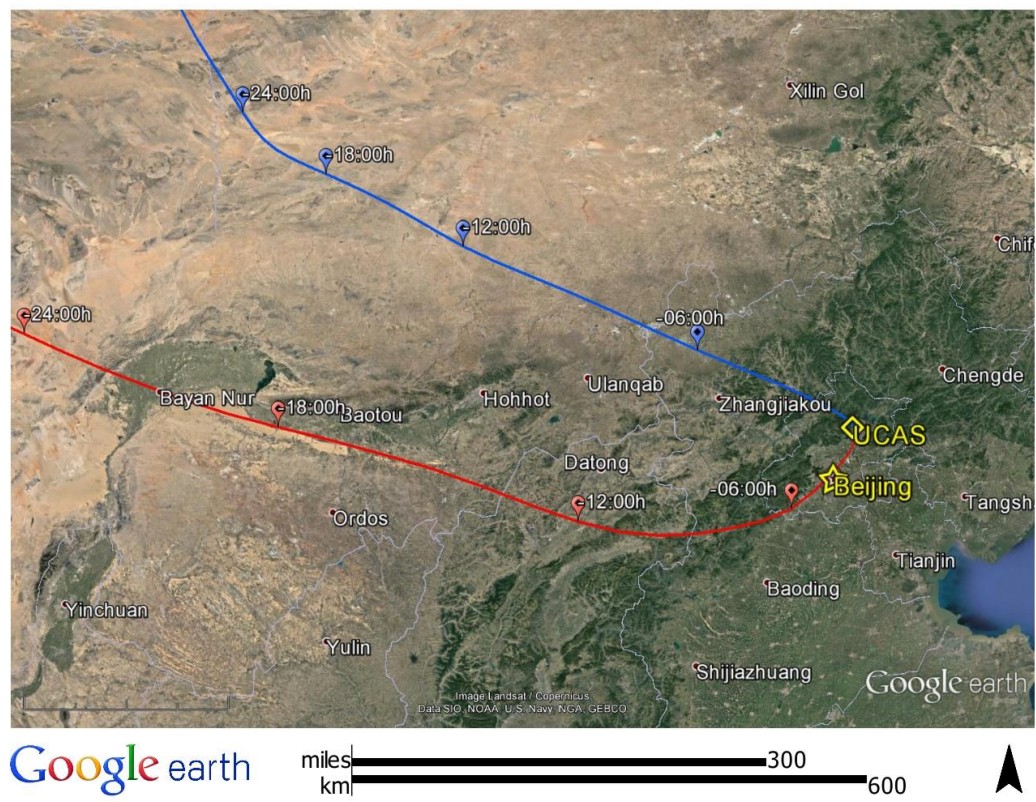

**Figure 1. NOAA HYSPLIT back trajectory analysis of air masses arriving at the field site. Two typical back trajectories were chosen to show cases for clean and pollution episodes. For the clean case, the example presents 08:00 CNST on 25**
5 **February, denoted by the blue line. For the pollution case, an example for 08:00 CNST 3 March is shown by the red line.**

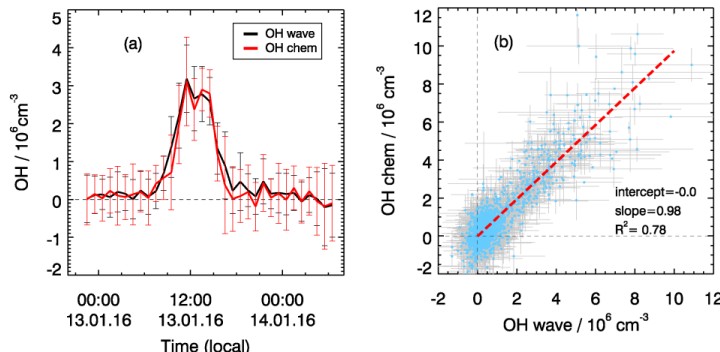

**Figure 2. Comparison of OH measurements using wavelength modulation (OH wave) and a chemical modulation reactor**
10 **(OH chem). (a) An example of chemical modulation test measurement from 13 to 14 January. Hourly averaged results are shown where the vertical bars denote variability. (b) Correlation plot between $OH_{wave}$ and $OH_{chem}$ for all chemical modulation experiments. Data points show 5 minutes averaged results with horizontal and vertical lines showing 1σ variability.**





**Figure 3. Time series of measured photolysis frequencies (j(O$^1$D), j(NO$_2$)), ambient temperature (T), particle mass concentrations with aerodynamic diameter below 2.5 um (PM$_{2.5}$), and concentrations of absolute water vapour (H$_2$O), CO, O$_3$, Ox (=O$_3$+NO$_2$), NO, NO$_2$, SO$_2$ and HONO. The grey areas denote nighttime. The labels in the first row denote the classification for background (B, green), clean (C, blue) and polluted (P, red) episodes.**



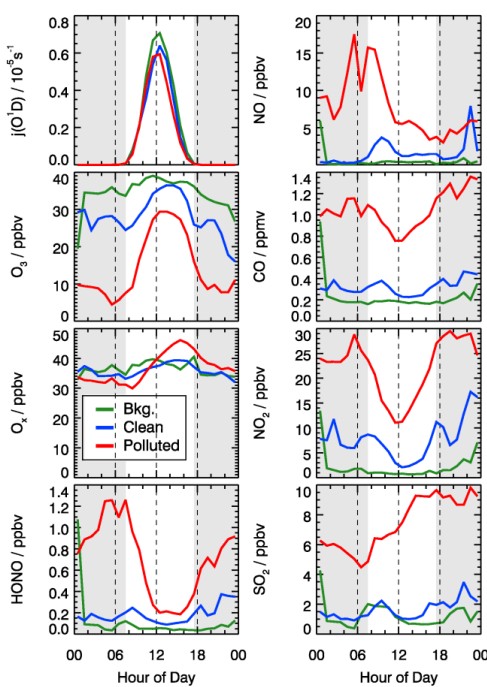

**Figure 4.** Mean diurnal profiles of measured photolysis frequencies ($j(O^1D)$ and concentrations of NO, CO, $O_3$, $O_x$ (=$O_3$+$NO_2$), $NO_2$, HONO and $SO_2$ for background (green), clean (blue) and polluted (red) episodes. The grey areas denote nighttime.



**Figure 5.** Time series of observed (red) and modeled (blue) OH, HO₂, the sum of RO₂, and of $k_{OH}$. The OH measurements were achieved by wavelength modulation and chemical modulation (CM is an abbreviation for chemical modulation, see text in section 2.2.1). The grey areas denote nighttime. The labels in the first row denote the classification for background (B, green), clean (C, blue) and polluted (P, red) episodes.



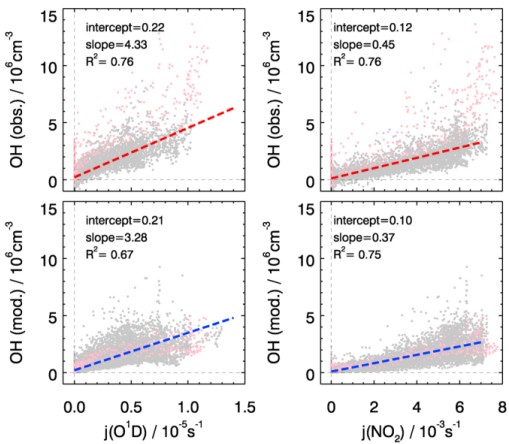

**Figure 6. The correlation between observed (upper panels) and modeled (lower panels) OH concentrations and photolysis frequencies of j(O$^1$D) (left panels) and j(NO$_2$) (right panels). The pink dots denote the results obtained on 27 and 28 February, which are not included in the correlation analysis. The units of the slope and the intercept are 10$^{11}$ cm$^{-3}$ s$^{-1}$ and 10$^6$ cm$^{-3}$, respectively.**

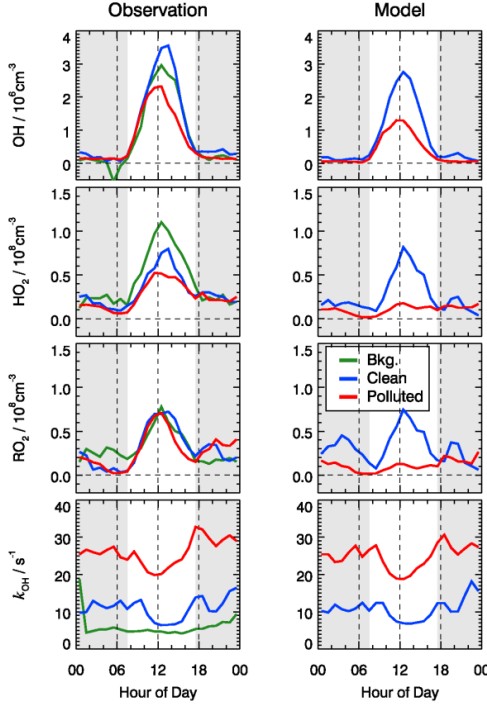

**Figure 7. Mean diurnal profiles of observed (left column) and modeled (right column) OH, HO$_2$, RO$_2$, and $k_{OH}$ for three different chemical and meteorological conditions. The category for background, clean and polluted episodes are the same as in Table 2, similar applied to Fig. 9 and Fig. 12. The grey areas denote nighttime.**





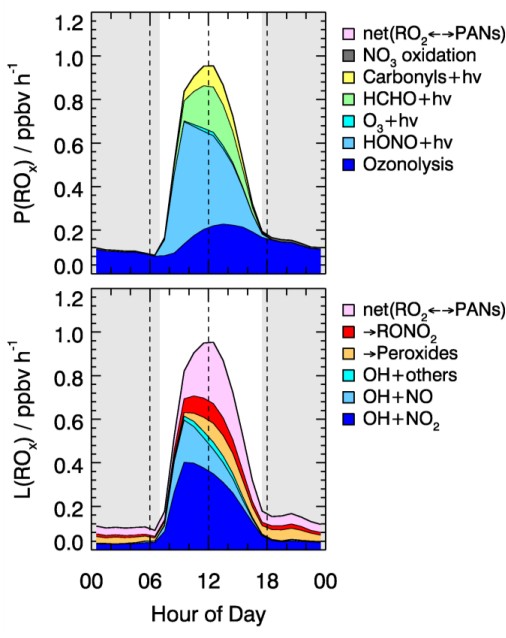

**Figure 8. Hourly averaged primary sources and sinks of RO$_x$ radicals derived from model calculations for all episodes.**
5 **The grey areas denote nighttime.**



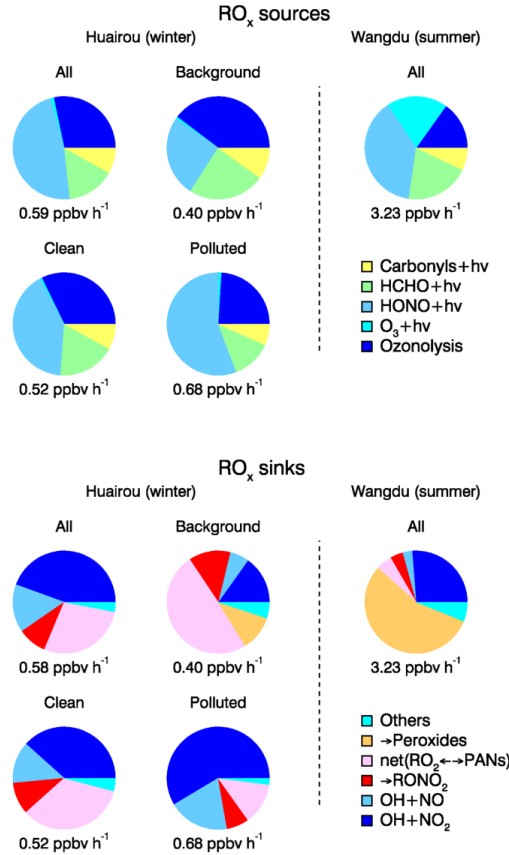

**Figure 9. Comparison of RO$_x$ primary sources and sinks from model calculations during different episodes for daytime averaged conditions. The budget analysis for the summer Wangdu campaign is plotted for comparison (Tan et al., 2017).**



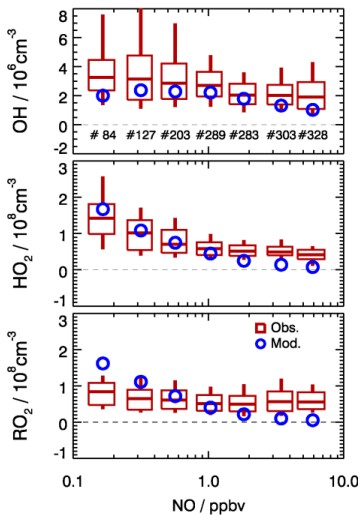

**Figure 10.** Dependence of observed and model-calculated of OH, HO$_2$ and RO$_2$ concentrations on NO concentrations for daytime conditions (j(O$^1$D) > 1.0×10$^{-6}$ s$^{-1}$). Boxes give the 75% and 25% percentiles, the center lines the median and vertical lines the 90% and 10% percentiles for each NO interval. Only median values are shown for model results. Numbers in the upper panel give the number of data points included in the analysis of each NO interval.

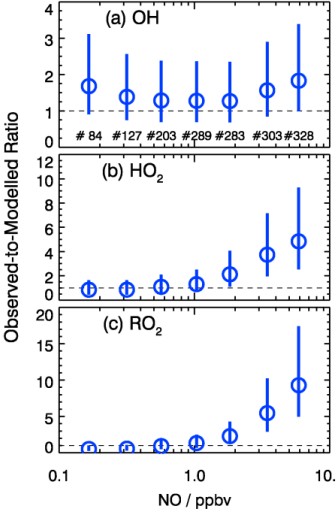

**Figure 11.** Dependence of observed-to-modeled ratio of OH, HO$_2$ and RO$_2$ on NO concentrations for daytime conditions (j(O$^1$D) > 1.0×10$^{-6}$ s$^{-1}$). The vertical lines denote the combined uncertainty from radical measurements and model calculations via error propagation. Numbers in the upper panel give the number of data points included in the analysis of each NO interval.




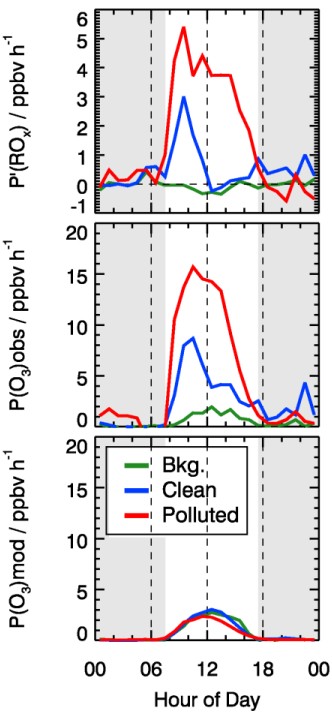

**Figure 12. Mean diurnal profiles of calculated missing RO$_x$ source (see text) and local ozone production determined from measured and modeled radical concentrations. The grey areas denote nighttime.**