# Peer review of "Wintertime photochemistry in Beijing: Observations of $RO_x$ radical concentrations in the North China Plain during the BEST-ONE campaign"

_Atmospheric Chemistry and Physics, 2018_

## Referee Comment (RC1) · Anonymous Referee #1 · 8 May 2018

This paper presents measurements of OH, HO2 and RO2 radicals during the winter (January – March) at a suburban site near Beijing. The measurements are separated according to three different chemical groups based on measured total OH reactivity, CO, and PM2.5, and were classified as "background," "clean," and "polluted." The radical concentrations were modeled using a 0D model based on the RACM2 condensed chemical mechanism constrained by the measured concentrations of NOx, VOCs, and others. The authors find that overall, the model can reproduce the observed OH to within 30%, with photolysis of HONO being the dominant OH source. However, the

agreement is better during "clean" episodes compared to "polluted" episodes, with the model underestimating the observed OH by a factor of 1.8 on "polluted" days. Similar to OH, the model is able to reproduce the observed $HO_2$ and $RO_2$ concentrations during "clean" episodes, but underestimates the observed $HO_2$ and $RO_2$ concentrations by a factor of 5 during "polluted" episodes, implying that the model is underestimating the instantaneous rate of ozone production during these periods.

Because the model was able to reproduce the observed OH reactivity, the authors conclude that the underestimation of $HO_2$ and $RO_2$ concentrations by the model is not due to unmeasured VOCs, but rather due to a missing primary source of peroxy radicals. Using a steady-state approach, the authors calculate a missing peroxy radical source rate of approximately 5 ppb/hour during "polluted" episodes. However, while addition of a generic peroxy radical source would improve the agreement between the model and the measured peroxy radical concentrations, addition of this source would increase the modeled OH concentrations by a factor of 5 greater than the measured concentrations, suggesting a significant missing OH sink, in contrast to the agreement of the measured and modeled OH reactivity. Overall, the authors find that the measured OH concentrations during this winter study were significantly greater than predicted by global models, suggesting that winter photochemistry in the region may be significantly greater than previously believed.

The paper addresses an important topic relevant to ACP. I recommend publication after the authors have addressed the following comments.

1) The authors conduct several chemical modulation tests to measure interferences using an "improved" chemical modulation reactor (CMR). Unfortunately there are limited details on the design of the CMR and how it was improved over the previous version. The paper would benefit from an expanded discussion of the CMR, including a schematic diagram, which could be included in a supplement.

2) The chemical modulation experiments should have allowed a direct measurement

of the ozone interference that is subtracted from the wavelength modulation signal (equation 2). Did the chemical modulation experiments confirm the correction for the ozone interference?

3) For the regression of the chemical modulation measurements versus the wavelength modulation measurements (Figure 2), the authors should clarify the regression method. They should use a bivariate regression weighted by the measurement precision of both OH chem and OH wave.

4) The description of the RO2 measurements appears to be incomplete, as the addition of NO and CO would result in the measurement of both RO2 + HO2 +OH (ROx as described in Fuchs et al., 2008). Measurements with CO addition only result in detection of HOx (OH + HO2) only. To obtain measurements of RO2 only, the measured HOx concentrations must be subtracted from the measured ROx concentrations. Based on the description in Tan et al. (2017), it appears that the HOx measurements from the other two axis are used to obtain the RO2 concentrations from the ROx measurements, but this should be clarified in this paper.

5) Related to this, it is not clear that the uncertainty for the RO2 measurements listed in Table 2 reflect the fact that the HOx measurements are subtracted from the ROx measurements. Fuchs et al. (2008) estimates the accuracy of the RO2 measurements to be approximately 20%. This should be clarified. In addition, the authors state that the measurement accuracies reflect both the "the uncertainty of the calibration source (10%, $1\sigma$) and the $1\sigma$ standard deviation of the variability of individual calibration sensitivities" (page 4). However the accuracies of the OH and HO2 measurements in Table 2 appear to only reflect the standard deviation of the variability of the individual calibration sensitivities, and do not appear to include the uncertainty associated with the calibration source. This should be clarified.

6) The authors should also provide an estimate of the model uncertainty.

7) Figure 7 could be improved to better show the model/measurement agreement/disagreement. Instead of separating the plots with measurements on one side and modeling on the other, I would suggest separating them by episode (background, clean, polluted), and then showing the model results and the measurements on the same plot, including the measurement and model variability, similar to that done in Tan et al. (2017).

8) While the measured/modeled ratios illustrated in Figure 11 suggest that the increase in the modeled underestimation of HO2 as a function of NO is similar to that observed previously (page 12), the HO2 measurements in some of the previous studies mentioned may have suffered from the RO2 interference discussed on page 3, resulting in reported HO2 measurements that may be greater than the actual HO2 concentrations. This potential interference would enhance the model-measurement discrepancies reported in these studies. This should be clarified in the discussion on page 12 of the manuscript.

---

## Referee Comment (RC2) · Anonymous Referee #2 · 20 May 2018

This paper reports in situ ground-based measurments of atmospheric radical species at a suburban site to the north of Beijing. State-of-the-science techniques are used to measure OH, HO2 and RO2 radicals, and the OH lifetime, augmented by established methods for other importance trace gas components. The potential for interferences in the OH measurements is carefully considered and could be dismissed.

A budget analysis derived from the observed data, and 0-D box model simulations,

is presented, the key result from which is that the model performed reasonably well (agreement within 50%) for "clean" days, but under more polluted conditions with higher NOx levels, the observed RO2 levels exceeded those modelled by up to a factor of 5, pointing to a significant discrepancy in understanding (or observations).

The subject of the paper is directly within the ACP remit, and the issues addressed are topical and at the forefront of our understanding. paper is broadly well written and clearly presented; the figures are of high quality. I have several minor comments, and three more significant points I would like the authors to consider in the ACPD Discussion :

Main comments

-While the measurements and analysis presented appear robust, the paper concludes that there is a significant discrepancy in quantitative understanding – but without assessing potential causes for this. I find this a little unsatisfactory – I would like the authors to add some suggestions – both related to measurement methods, model limitations and potential new chemical understanding – which could resolve these. These may include suggestions for future work to move the situation forwards.

-Is the site location representative of Beijing ? Huairou is on the northern perimeter of Beijing, adjacent to the higher ground to the north and outside of much of the city development. Is the chemical environment then representative of "downtown" Beijing, in the city centre. The authors should confirm this (eg through comparison of basic AQ metrics), and I would like them to reflect the presentation of their location as different from (e.g.) the CARE campaigns etc in the text, and potentially the manuscript title

-Accumulating evidence is pointing to Cl chemistry being important for radical formation; ClNO2 observations were not made during this campaign – what is the sensitivity of the conclusions to the assumed ClNO2 / Cl atom levels – how might this (and the NOx-dependence of the availability of Cl vs inorganic reservoir formation) affect the radical budgets ?

Other Comments

p. 1Line 34 – "series of control provisions" give dates to increase the relevance of the paper in future years

P2 L1 – national trends needs a reference

P2 L7 "attack" ?

P2 L12 "guarantees" is a bold word to use – what about species with slow OH reaction (eg CH4)

P2 L14 O3 photolysis is not the dominant source of HOx in the BL, outside of remote marine environments – as shown by eg fig 9 of this paper

P2 L 18 "general expectation" – what about previous measurements of OH

P2 L25+ Needs reference to Hofzumahaus et al. Science paper

P3 L10 Compare NOx PM etc with central Beijing to justify site description as "Beijing"

P5 L10+ The description of the checks of OH-chem vs OH-wave is good and reassuring, but more detail on the RO2 and HO2 method and particularly the uncertainties in these would be useful

P6 L5 Does the LOPAP discrepancy correlate with other factors – eg aerosol nitrite levels ?

P6 L30 How did the model constraint work. Was the model simply updated to the observed levels every ?15 mins – does this introduce noise into the output concentrations. What about spin-up time to simulate intermediate species.

P7 L14 Give values for the thresholds used to define the pollution regimes

P9 L12 jO1D and jNO2 should not be correlated given the different adsorption spectra and quantum yield wavelength dependence for O3-O1D and NO2 photolysis. See discussion in Rohrer & Berresheim, Nature 2006 and other HOx measurement / j correlation analyses eg Smith et al. ACP 2006

P9 L38 – can you expand on the "NO measurement artefact" ?

P11 L18 reword

P13 L33 – Need to be clear that the observations and model only determine the local chemical ozone production rate, while a wider view (Eularian or trajectory) is needed to compare with O3 levels (ie accounting for advection). Also relevant to Fig 12b. -Modelling – how significant were modelled VOC degradation products in terms of increasing the OH reactivity, compared with the measured (parent) VOCs ?

P15 L35 – see main comment above

---

## Author Comment (AC1) · 28 Jun 2018

Please find the response in the supplement. We also add 3 more figures in the supplement as supporting materials.

Please also note the supplement to this comment:
https://www.atmos-chem-phys-discuss.net/acp-2018-359/acp-2018-359-AC1-supplement.zip

---

## Author Comment (AC2) · 28 Jun 2018

Please find the response in the supplement. We also add 3 more figures in the supplement as supporting materials (same as the reply to reviewer #1).

Please also note the supplement to this comment:
https://www.atmos-chem-phys-discuss.net/acp-2018-359/acp-2018-359-AC2-supplement.zip

---

## Author Response (AR1)

We would like to thank the reviewer for comments and questions which helped us to improve the manuscript. The reviewer comments are given below together with our responses and changes made to the manuscript.

1) The authors conduct several chemical modulation tests to measure interferences using an "improved" chemical modulation reactor (CMR). Unfortunately there are limited details on the design of the CMR and how it was improved over the previous version. The paper would benefit from an expanded discussion of the CMR, including a schematic diagram, which could be included in a supplement.

**Answers:**

We added a schematic plot of the CMR in the supplement. In this version, we improve the mixing of ambient air with an added agent by using a two-needle injector system.

We changed the text on Page 4 Line 26-30 to be "In the present study, an improved CMR device was used for some selected time periods during clean and polluted air conditions. The device consisted of a Teflon tube with an inner diameter of 1.0 cm and a length of 8.3 cm (Fig. S2). About 20 slpm of ambient air was drawn through the tube by a blower, of which 1 slpm was sampled into the OH detection cell. In the current design, two small stainless steel tubes (outer diameter 1/16 inches) were arranged at the entrance of the Teflon tube opposite to each other (compared to one injector in the previous version described in Tan et al. 2017)). This change of the injector system design could improve the mixing of ambient air with the injected propane."

2) The chemical modulation experiments should have allowed a direct measurement of the ozone interference that is subtracted from the wavelength modulation signal (equation 2). Did the chemical modulation experiments confirm the correction for the ozone interference?

**Answers:**

We added a sentence on Page 5 Line 5 that "The ozone photolysis is a known interference in the OH measurement using wavelength modulation (Holland et al. 2003). This interference was characterized in laboratory experiments and subtracted in $OH_{WM}$. The background signal determined in the chemical modulation contains the information about interferences including

that from ozone. This signal is consistent with the correction applied to the $OH_{WM}$ detection scheme. This can be seen, for example, in the good agreement between $OH_{WM}$ and $OH_{CM}$ during nighttime, when the ozone interference was a large fraction of the uncorrected $OH_{WM}$ signal. "

3) For the regression of the chemical modulation measurements versus the wavelength modulation measurements (Figure 2), the authors should clarify the regression method. They should use a bivariate regression weighted by the measurement precision of both OH chem and OH wave.

**Answers:**

The regression is done with a bivariate weighted method and the figure 2b is updated accordingly, which resulted in slightly different from the origin one (polyfit). We changed the sentence on Page 5 Line 14-16 "The regression is done using a bivariate regression weighted by the measurement errors of both signals, $OH_{CM}$ and $OH_{WM}$. The slope of this correlation is close to unity (1.1) for the various encountered chemical conditions. Small intercept ($0.2 \times 10^6$ $cm^{-3}$ smaller than the detection limit) is found indicating that no significant bias in the low concentration range…"

4) The description of the RO2 measurements appears to be incomplete, as the addition of NO and CO would result in the measurement of both RO2 + HO2 +OH (ROx as described in Fuchs et al., 2008). Measurements with CO addition only result in detection of HOx (OH + HO2) only. To obtain measurements of RO2 only, the measured HOx concentrations must be subtracted from the measured ROx concentrations. Based on the description in Tan et al. (2017), it appears that the HOx measurements from the other two axes are used to obtain the RO2 concentrations from the ROx measurements, but this should be clarified in this paper.

**Answers:**

We added a sentence in Page 4 Line 5 that "The measurements from the other two fluorescence cells are used to calculate the contributions from OH and $HO_2$ and subtracted to retrieve the $RO_2$ measurements."

The same subtraction is applied to the HO2 measurement. So we added a sentence on Page 3 Line 28 "The contribution of OH is subtracted using the measurement in the OH channel and OH sensitivity in the $HO_2$ channel."

5) Related to this, it is not clear that the uncertainty for the RO2 measurements listed in Table 2 reflect the fact that the HOx measurements are subtracted from the ROx measurements. Fuchs et al. (2008) estimates the accuracy of the RO2 measurements to be approximately 20%. This should be clarified. In addition, the authors state that the measurement accuracies reflect both the "the uncertainty of the calibration source (10%, 1σ) and the 1 σ standard deviation of the variability of individual calibration sensitivities" (page 4). However the accuracies of the OH and HO2 measurements in Table 2 appear to only reflect the standard deviation of the variability of the individual calibration sensitivities, and do not appear to include the uncertainty associated with the calibration source. This should be clarified.

**Answers:**

We changed the sentence in Page 4 Line 9-11 to be "The accuracies include the uncertainty of the calibration source (10%, 1σ) and the 1σ standard deviation of the variability of individual calibration sensitivities (OH: 10%, HO2: 13%, RO2: 11%). The accuracies are calculated from Gaussian error propagation 14%, 17% and 23% for OH, $HO_2$, and $RO_2$, respectively." We also changed the numbers in Table 1 accordingly.

6) The authors should also provide an estimate of the model uncertainty.

**Answers:**

We added a sentence in Page 7 Line 2 "The uncertainty of the model calculations is derived from the uncertainties in the measurements used as model constraints and the reaction rate constants. Taking into account the uncertainties of both measurements and kinetic rate constants, a series of tests based on Monte Carlo simulations show that the 1σ uncertainty of the model calculations is approximately 40% (Tan et al. 2017)."

7) Figure 7 could be improved to better show the model/measurement agreement/disagreement. Instead of separating the plots with measurements on one side and modeling on the other, I would suggest separating them by episode (background, clean, polluted), and then showing the model results and the measurements on the same plot, including the measurement and model variability, similar to that done in Tan et al. (2017).

**Answers:**

In Figure 7, we try to emphasize that the observed radical concentrations are similar in all three cases while the model predicts radical concentrations that are significantly decreased in the polluted periods compared to the clean episodes. The measurement-model discrepancy is highlighted in the NO dependence (Figure 10 and 11). However, we agree that this plot is also insightful and thus we show it in the supplement.

We added a sentence on Page 9 Line 34 "In fact, the observed radical concentrations are rather comparable in all episodes, while the model predicts a suppression of radical concentrations in the polluted episodes. This is most obviously seen for $HO_2$ and $RO_2$. The comparison between observed and modelled OH, $HO_2$, and $RO_2$ concentrations for clean and polluted episodes are shown in Fig. S3."

8) While the measured/modeled ratios illustrated in Figure 11 suggest that the increase in the modeled underestimation of HO2 as a function of NO is similar to that observed previously (page 12), the HO2 measurements in some of the previous studies mentioned may have suffered from the RO2 interference discussed on page 3, resulting in reported HO2 measurements that may be greater than the actual HO2 concentrations. This potential interference would enhance the model-measurement discrepancies reported in these studies. This should be clarified in the discussion on page 12 of the manuscript.

**Answers:**

We added a paragraph on Page 12 Line 13 "$HO_2$ measurements in previous field campaigns could have suffered from interferences from specific $RO_2$ species, so that the reported observed-to-model ratios could have been even larger in these campaigns."

**Anonymous Referee #2**

We would like to thank the reviewer for comments and suggestion which helped us to improve the manuscript. The reviewer comments are given below together with our responses and changes made to the manuscript.

**Main comments**

1. While the measurements and analysis presented appear robust, the paper concludes that there is a significant discrepancy in quantitative understanding – but without assessing potential causes for this. I find this a little unsatisfactory – I would like the authors to add some suggestions – both related to measurement methods, model limitations and potential new chemical understanding – which could resolve these. These may include suggestions for future work to move the situation forwards.

**Answers:**

We added a paragraph on Page 13 Line 26 following the discussion of the role of Chlorine chemistry: "The underestimation of $RO_2$ concentrations in the model occurred mainly during the pollution episodes when the measurement site was influenced by air mass transported from the Beijing central area or by local emissions. Since $ClNO_2$ and molecular chlorine were not measured in this campaign, their possible role in the production of $RO_x$ is difficult to quantify. High $N_2O_5$ concentrations were observed in this campaign with values up to 10 ppbv during the pollution episodes (Wang et al., 2017a) and also aerosol chlorine was abundant (up to 7 μg/m$^3$) to facilitate the production of $ClNO_2$. Therefore, the $ClNO_2$ has the potential to explain, at least part of the missing $RO_2$ source. However, the production rate of $ClNO_2$ depends on the $N_2O_5$ aerosol uptake coefficient and the $ClNO_2$ yield, both of which can be highly variable (Tham et al., (2018)). Therefore, measurements of chlorine chemistry related species would be essential to evaluate its effect on the $OH$-$HO_2$-$RO_2$ radical system, but they are not available here."

In the summary, we added a sentence on Page 15 Line 37 "Although the chlorine chemistry has the potential to partly explain the missing radical source, its effect on radical concentrations could not be quantified due to the lack of $ClNO_2$ measurements. In the future, the measurements of chlorine-related species (e.g. $ClNO_2$, $Cl_2$) would be helpful to gain more insights what the contributions of $ClNO_2$ are to the radical sources and to the formation of secondary pollution. "

2. Is the site location representative of Beijing? Huairou is on the northern perimeter of Beijing, adjacent to the higher ground to the north and outside of much of the city development. Is the chemical environment then representative of "downtown" Beijing, in the city centre. The authors should confirm this (e.g. through comparison of basic AQ metrics), and I would like them to

reflect the presentation of their location as different from (e.g.) the CARE campaigns etc in the text, and potentially the manuscript title
**Answers:**

We added the measurement of CO, $NO_2$, $O_3$, $SO_2$, and $PM_{2.5}$ obtained at 12 stations in Beijing downtown as well as the measurement at the campaign site in the supplement (Figure S1). We found a consistent trend in these measurements.

We added on Page 3 Line 10 "As shown in Fig. S1, the concentrations of CO, $NO_2$, $SO_2$, and $PM_{2.5}$ observed at the site showed good correlations with the measurements conducted in the city center (12 EPA stations), although the concentrations were in the lower range during pollution episodes. However, the city center measurements showed higher peak values with large variability, which were mainly caused by local emission. This demonstrates that the measurement site is a representative for conditions in the Beijing with minor influence by local emission. In contrast, a previous field campaign that included $OH-HO_2-RO_2$ radicals measurements were performed at a rural site in Wangdu in summer 2014 (Tan et al., 2017). The Wangdu site was located in the middle of the North China Plain, about 200 km southwest of Beijing. The site was mainly influenced by regional transportation of air pollutants from anthropogenic emissions (Fuchs et al., 2017)."

3. Accumulating evidence is pointing to Cl chemistry being important for radical formation; ClNO2 observations were not made during this campaign – what is the sensitivity of the conclusions to the assumed ClNO2 / Cl atom levels – how might this (and the NOx-dependence of the availability of Cl vs inorganic reservoir formation) affect the radical budgets?
**Answers:**

Please refer to the answer to the first comment.

**Other Comments**
p. 1Line 34 – "series of control provisions" give dates to increase the relevance of the paper in future years
**Answers:**

We revised the sentence as "After a series of air pollution control provisions have been implemented by the Chinese government in the last 15 years to improve the air quality in China. "

P2 L1 – national trends need a reference
**Answers:**

We added a reference from Kan et al. (2012).

Kan, H., Chen, R., and Tong, S.: Ambient air pollution, climate change, and population health in China, Environ Int, 42, 10-19, https://doi.org/10.1016/j.envint.2011.03.003, 2012.

P2 L7 "attack"?
**Answers:**

We revised the word as "initiate the oxidation of".

P2 L12 "guarantees" is a bold word to use – what about species with slow OH reaction (eg CH4)
**Answers:**

We revised the word as "facilitate".

P2 L14 O3 photolysis is not the dominant source of HOx in the BL, outside of remote marine environments – as shown by eg fig 9 of this paper
**Answers:**

We changed the sentence as "In wintertime, the radical chemistry is less active than in summertime because the solar radiation is weaker due to the higher solar zenith angle. For example, one of the important OH primary sources, photolysis of ozone, is strongly reduced by the smaller photolysis rate and the lower water vapor abundances at low temperatures during wintertime."

P2 L 18 "general expectation" – what about previous measurements of OH
**Answers:**

We changed the sentence as "… The significant difference between OH concentrations in summer- and wintertime indicates that the radical chemistry only plays a minor role in winter. Especially during particle pollution events, the dimming effect of aerosol will further attenuate the solar radiation and thus lowering the radical chemistry activity."

P2 L25+ Needs reference to Hofzumahaus et al. Science paper
**Answers:**

Reference is added.

P3 L10 Compare NOx PM etc with central Beijing to justify site description as "Beijing"

**Answers:**

Please refer to answer to main comments 2.

**Answers:**

We expanded the description of HO2 measurements on Page 3: "$HO_2$ was converted to OH by NO addition below the sample nozzle in a second fluorescence cell that had otherwise the same design as the OH cell. The contribution of OH is subtracted using the measurement in the OH channel and OH sensitivity in the $HO_2$ channel. It is known that the measurement of $HO_2$ by chemical conversion can introduce interference from specific $RO_2$ radicals (Fuchs et al., 2011; Whalley et al., 2013; M. Lew et al., 2018). The best way to reduce the interference is to decrease the NO mixing ratio in the $HO_2$ cell (Fuchs et al., 2011; Whalley et al., 2013; Whalley et al., 2017; Tan et al., 2017). In this study, the NO addition was reduced to minimize the $RO_2$ interference without losing too much of the $HO_2$ conversion. The NO concentration was switched every two minutes between the additions of 0.5 standard millilitres per minute (sccm) and of 2 sccm from a mixture of 1% NO in $N_2$. This yields a nominal mixing ratio of 2.5 ppmv and 10 ppmv of NO in a sample flow of 1 SLM and Especially flow of 1 SLM. No significant difference was found for the two $HO_2$ data sets showing that the $HO_2$ measurements were interference-free."

We expanded the description of RO2 measurements on Page 4: "$RO_2$ measurements with the LIF instrument require the conversion of $RO_2$ to OH. This was done in two steps. In a reaction flow tube (pressure 25 hPa, volume 2.8 L, sample flow 7.5 SLM) high concentrations of CO (1100 ppm) and NO (0.7 ppmv) were added to convert $RO_x$ to $HO_2$ (Fuchs et al., 2008). CO was added to suppress the conversion from $HO_2$ to OH to avoid wall losses of OH radicals in the reactor. The $HO_2$ radicals were transferred to the fluorescence cell through a nozzle pinhole with a diameter of 4.0 mm. The fluorescence cell was operated at 4 hPa like the other fluorescence cells. The $HO_2$ radicals were finally converted to OH radicals using a flow of 5 sccm of pure NO yielding a nominal mixing ratio of 1100 ppm NO within a sample flow of 3.5 SLM and an $N_2$-sheath flow of 1 SLM. The measurements from the other two fluorescence cells were used to calculate the contributions from OH and $HO_2$ and subtracted to retrieve the $RO_2$ measurements."

**Answers:**

Unfortunately, the aerosol nitrite was not measured in this campaign. We tried to correlate the discrepancy between the two LOPAP measurements with measured parameters but could not identify a parameter that correlates with the observed differences in the two measurements. Since the cause of the LOPAP discrepancy is not clear yet, the discrepancy adds to the uncertainty of the measurement.

P6 L30 How did the model constraint work. Was the model simply updated to the observed levels every 15 mins?– does this introduce noise into the output concentrations. What about spin-up time to simulate intermediate species.
**Answers:**

We added a few sentences on Page 6 Line 33: "The model was operated in a time-dependence mode with 5-min time resolution for which measurements used as constraints updated the model values. 2 days spin-up time was used to initiate the model."

P7 L14 Give values for the thresholds used to define the pollution regimes
**Answers:**

We changed the sentence in Page 7 Line 14-16 as "The measured OH reactivity was used to separate polluted from clean periods by a threshold value of $k_{OH}$ = 15 s$^{-1}$ (daily average). This corresponds also to CO mixing ratios higher than 1ppmv and PM$_{2.5}$ higher than 50µg/m$^3$ since these parameters were highly correlated."

P9 L12 jO1D and jNO2 should not be correlated given the different adsorption spectra and quantum yield wavelength dependence for O3-O1D and NO2 photolysis. See discussion in Rohrer & Berresheim, Nature 2006 and other HOx measurement / j correlation analyses eg Smith et al. ACP 2006 P9
**Answers:**

Although the absorption spectra and quantum yield wavelength dependence for $O_3$-$O^1D$ and $NO_2$ photolysis are different, the relative good correlation between j($O^1D$) and j($NO_2$) was found in this campaign ($R^2$=0.87). Two kinds of correlation can be found. (1) If the photolysis frequencies are changed by clouds, they are linearly correlated. (2) If they are changed by different solar zenith angles, they have a square root dependence. The actual exponent varies between 0.5 and 1, depending on the on the local meteorological conditions during a certain campaign at a certain location.

L38 – can you expand on the "NO measurement artefact" ?

**Answers:**

It is not really an artefact but just because NO concentration was below detection limit (60 pptv). If NO is below LOD, the measurement will show a large variability which resulted in a large fluctuation in the model. Therefore, we cancel the word artefact here.

We changed the sentence in Page 10 Line 1-6 to "…which could be the result of a NO measurement below the detection limit. The observed NO concentrations were often below the limit of detection of the $NO_x$ instrument (60 pptv), which did not allow precise measurements due to the fluctuation of the background signal. Besides, it also led to large variability in the modelled $NO_3$ and result in overprediction in the nighttime $RO_2$ (Tan et al., 2017). On the other hand, a small bias …"

P11 L18 reword
**Answers:**

We changed the sentence to "During the summertime in Wangdu, the contribution from alkene ozonolysis was only 15% (Fig. 9). However, the absolute rate was 0.47 ppbv/h, larger than what was observed in Huairou/Beijing in wintertime (0.16 ppbv/h)."

P13 L33 – Need to be clear that the observations and model only determine the local chemical ozone production rate, while a wider view (Eularian or trajectory) is needed to compare with O3 levels (ie accounting for advection). Also relevant to Fig 12b.
**Answers:**

We changed the sentence as "This indicates that the $O_x$ was produced by local photochemical reactions and/or was transported from the upwind areas. The change of the $O_x$ concentration is caused by both chemical production/destruction and physical processes (advection, vertical mixing, deposition and so on). Therefore, it is important to note that the large local production rate is not necessarily observed in the measured, local $O_x$ concentration. In this study, we compared the chemical production rate and the $O_x$ concentration change to illustrate whether chemical production can support the $O_x$ concentration increase."

Modelling – how significant were modelled VOC degradation products in terms of increasing the OH reactivity, compared with the measured (parent) VOCs?
**Answers:**

We changed the sentence in Page10 Line 11-16 as "The model was capable to reproduce the directly observed OH reactivity within 10% during all episodes (Fig. 7). This calculation includes the reactivity from observed VOCs (about 73-83% of the observed $k_{OH}$) and the estimated contributions from OVOCs calculated by the model (17-27%). The speciation of the total OH reactivity showed that the major OH reactants were $NO_x$ and CO. On average, CO and $NO_x$

contributed 23% and 37% to the total OH reactivity, respectively. 18% of the observed reactivity can attribute to measured VOC species. In comparison, the model generated species contributed 22% to the total reactivity. For the polluted episodes, the average OH reactivity increased from 10 to 26 s$^{-1}$ with a significant increase in the relative contributions from the inorganic compounds (from 52% to 63%)."

P15 L35 – see main comment above
**Answers:**

Please refers to the answer to main comments 1 and 3.

[revised manuscript text omitted]

---

## Author Response (AR2)

We thank the reviewer for comments.

However, one addition to the manuscript that is not clear relates to the correction of the HO2 measurements using the measured OH concentration from the OH detection axis on page 4, line 7. The authors state that "The contribution of OH is subtracted using the measurement in the OH channel and OH sensitivity in the HO2 channel." This is unnecessarily confusing as it seems to imply that the OH sensitivity in the HO2 channel is used to determine OH concentrations in the OH channel. I believe what the authors are stating is that the contribution of OH to the HO2 measurements in the HO2 channel are subtracted using the measurements from the OH channel after accounting for the different sensitivities in the two channels, similar to the correction of ROx measurements to obtain RO2 concentrations (page 4, line 23).

**Answer:** We revised the sentence on Page 4 Line 7 to be "To obtain the concentration of $HO_2$, the contribution of OH to the signal in the $HO_2$ cell is subtracted using the measurements of the OH cell after accounting for the different sensitivities in the two cells."

Related to this, it would be useful to specify the HO2 conversion efficiency using the two different NO flows (page 4), and whether the stated uncertainty in the HO2 measurements relates to both NO flow measurements.

**Answer:** We added sentence to specify the HO2 measurement in Page 4 Line 14 "This yields a nominal mixing ratio of 2.5 ppmv and 10 ppmv of NO in a sample flow of 1 SLM and an $N_2$-sheath flow of 1 SLM resulting in the $HO_2$ conversion efficiencies being 4% and 18%, respectively. No significant difference was found for the two $HO_2$ data sets showing that the $HO_2$ measurements were interference-free. Since the data interpretation in this study is in 5 min time resolution, both $HO_2$ measurements are averaged for 5 min time intervals. The accuracy and precision refer to the 5 min averaged data set (Table 1)."